# Retrieving monthly and interannual pH$_T$ on the East China Sea shelf using an artificial neural network: ANN-pH$_T$-v1

Xiaoshuang Li[1, 2], Richard Bellerby[1, 2], Jianzhong Ge[1], Philip Wallhead[2], Jing Liu[1], and Anqiang Yang[1]

[1]State Key Laboratory of Estuarine and Coastal Research, East China Normal University, Shanghai, 200241, China
[2]Norwegian Institute for Water Research, Bergen, 5006, Norway

*Correspondence to*: Richard Bellerby (Richard.Bellerby@niva.no)

**Abstract.** While our understanding of pH dynamics has strongly progressed for open ocean regions, for marginal seas such as the East China Sea (ECS) shelf progress has been constrained by limited observations and complex interactions between biological, physical, and chemical processes. Seawater pH is a very valuable oceanographic variable but not always measured using high quality instrumentation and according to standard practices. In order to predict total scale pH (pH$_T$) and enhance our understanding of the seasonal variability of pH$_T$ on the ECS shelf, an artificial neural network (ANN) model was developed using 11 cruise datasets from 2013 to 2017 with coincident observations of pH$_T$, temperature (T), salinity (S), dissolved oxygen (DO), nitrate (N), phosphate (P) and silicate (Si) together with sampling position and time. The reliability of the ANN model was evaluated using independent observations from 3 cruises in 2018, and showed a root mean square error accuracy of 0.04. The ANN model responded to T and DO errors in a positive way, S errors in a negative way, and the ANN model was most sensitive to S errors, followed by DO and T errors. Monthly water column pH$_T$ for the period 2000-2016 was retrieved using T, S, DO, N, P, and Si from the Changjiang Biology Finite-Volume Coastal Ocean Model (FVCOM). The agreement is good here in winter, while the reduced performance in summer can be attributed in large part to limitations of the Changjiang Biology FVCOM in simulating summertime input variables.

## 1 Introduction

Atmospheric carbon dioxide (CO$_2$) levels have increased by nearly 46%, from approximately 278 ppm (parts per million) in 1750 (Ciais et al., 2013) to 405 ppm in 2017 (Le Quéré et al., 2018). The oceans have absorbed approximately 48% of the anthropogenic CO$_2$ emissions (Sabine et al., 2004), resulting in decreasing long-term pH trends of ~0.02 decade$^{-1}$ in open ocean waters (e.g., Dore et al., 2009; González-Dávila et al., 2010; Bates et al., 2014; Lauvset et al., 2015). While a gradual decrease in pH is a predictable open ocean response to elevated anthropogenic CO$_2$ emissions, the seasonal changes and long-term trends in pH in coastal seas have not been fully understood due to the lack of long-term pH data and complexity of coastal systems. In this context, the development of approaches to predict carbonate chemistry parameters in coastal regions may assist both the management of local water quality and our wider understanding of the ocean carbon cycle.

Many attempts have been made to predict seawater pH by developing empirical relationships between pH and environmental variables, such as temperature (T) (Juranek et al., 2011), salinity (S) (Williams et al., 2016), dissolved oxygen (DO) (e.g., Juranek et al., 2011; Sauzède et al., 2017), nutrients (e.g., Williams et al., 2016; Carter et al., 2016, 2018), and longitude, latitude (Sauzède et al., 2017). Compared with traditional empirical methods, artificial neural networks (ANNs) have been proposed as powerful tools for modelling uncertain and complex systems such as ecosystems and environmental assessment (e.g., Olden and Jackson, 2002; Olden et al., 2004; Uusitalo, 2007; Raitsos et al., 2008; Chen et al., 2017). Their main advantage compared with e.g. multiple linear regression (MLR) models may be a greater flexibility and versatility in modelling complex nonlinear relationships. ANNs have been used for the retrieval of the partial pressure of carbon dioxide (pCO$_2$) (e.g., Friedrich and Oschlies, 2009; Laruelle et al., 2017), total alkalinity (e.g., Velo et al., 2013; Bostock et al., 2013; Sasse et al., 2013), total dissolved inorganic carbon (e.g., Bostock et al., 2013; Sasse et al., 2013), and phytoplankton functional types (e.g., Raitsos et al., 2008; Palacz et al., 2013). However, these studies mainly focus on the open ocean; relatively few studies have focused on

coastal seas, perhaps because of the complexity and heterogeneity of the continental shelves. Alin et al. (2012) developed an MLR model to reconstruct pH in the southern California Current System, while Moore-Maley et al. (2016) evaluated the interannual variability of near-surface pH using a one dimensional, biophysical, mixing layer model in the Strait of Georgia. To our knowledge, no empirical relationship for pH has yet been established for the ECS.

The ECS is the largest marginal sea in the western North Pacific Ocean and receives massive terrestrial inputs from the

Changjiang (Yangtze River). The shelf shallower than 200 m covers more than 70% of the entire ECS (e.g., Ichikawa and Beardsley, 2002; Lie and Cho, 2016), where the dominant currents present seasonal circulation patterns. The spatial and temporal distributions of the carbonate system have been investigated in the ECS (e.g., Chou et al., 2009; Cao et al., 2011; Qu et al., 2015), and were found to largely reflect the distributions of various water masses. The pattern of carbon sources and sinks exhibits substantial seasonal variation (Guo et al., 2015), and the ECS is generally considered as a sink of atmospheric

$CO_2$ throughout the year except in fall (e.g., Shim et al., 2007; Zhai and Dai, 2009). A mechanistic semi-analytical algorithm (MeSAA) was developed to study $pCO_2$ variations in response to various controlling mechanisms during summertime (Bai et al., 2015). However, the seasonal variability of pH has been very little studied in the ECS, mainly due to the limited observational coverage and irregular variability caused by seasonal fluctuations of the Changjiang discharge and anthropogenic processes. Developing methods to extend the seasonal coverage of pH data may thus help to improve our understanding of the

ocean carbon cycle in the ECS.

This paper is structured as follows: section 2 describes the cruise data and ANN model building; section 3 shows the performance, sensitivity and application of the ANN model. Summary and conclusions are summarized in the last section.

## 2 Data and method

### 2.1 Data

Ten cruises were conducted on the ECS shelf during the "Fund Committee Innovation Group Project" (Y22323101B) from 2013 to 2017 (Fig. 1), the summer cruise from 17 to 28 August 2013, 10 to 17 July 2014, 9 to 20 July 2015, 4 to 28 July 2016, 20 to 30 July 2017, the winter cruise from 21 to 28 February 2014, 15 to 28 February 2017, the spring cruise from 4 to 20 March 2013, 11 to 21 March 2015, 7 to 19 March 2016. T and S profiles were obtained directly using a conductivity temperature-depth/pressure (CTD) recorders (SBE 25plus or 911plus). Measurement of DO followed the Winkler procedure,

as described previously by Zhai et al. (2014). Nutrients samples were first filtered with 0.45 μm Whatman GF/F membrane, then stored in 250 mL HDPE bottles until chemical analysis. Nitrate (N), phosphate (P) and silicate (Si) were determined using a segmented flow analyzer (Model: Skalar SAN$^{PLUS}$, Netherlands) with a precision < 5% (Zhang et al., 2007), the detection limits are 0.14 μM for N, 0.06 μM for P, and 0.07 μM for Si. pH samples were stored in 140 mL brown borosilicate glass bottles and sterilized by addition of 50 μL saturated $HgCl_2$ solution. Three traceable pH buffers were used including NIST

(National Institute of Standards and Technology) buffers pH = 4.00, 7.02, 10.09. As described by Zhai et al. (2012, 2014), we converted it into total scale $pH_T$ by subtracting 0.143 and the overall accuracy of the $pH_T$ dataset was estimated as 0.01.

Three cruises were carried out on the ECS shelf in 2018 (Fig. 2) during the "National Natural Science Foundation Shared Voyage Plan", from 10 to 19 March, 12 to 20 July, 12 to 21 October, and one cruise was carried out near the Changjiang Estuary during May 2017 (Fig. 1). The measurement methods of T, S, DO, and nutrients are the same as that of the above ten

voyages. pH samples were stored in 500 mL high-quality borosilicate glass bottles without filtering and sterilized by addition of 200 μL saturated $HgCl_2$ solution until measurement in the lab. The $pH_T$ was measured at the temperature in the flow cell using an Automated Flow-through system for Embedded Spectrophotometry (AFtes) with a precision of 0.0005 pH unit and uncertainty of < 0.003 (Reggiani et al., 2016). Water samples were collected at three or four different depths during all cruises. We omitted data points where one or more other physical variables were missing. The three cruises during 2018 (Fig. 2) were

used to estimate model predicted performance as an exploratory dataset, while the remaining eleven cruises (Fig. 1) were used

to train the model as a confirmatory dataset. The final number of observations in the confirmatory dataset was 1854 (see Table 1 for more detailed information on the field survey).

## 2.2 Artificial neural network development

The ANN we used is a feed-forward multilayer perceptron (Tamura and Tateishi, 1997) with two hidden layers. The neurons
of each layer are connected with the neurons of the previous layer and the next layer by weights (Fig. 3a). The coefficients of the weight matrix are iteratively tuned in the training step. In order to avoid overfitting, a ten-fold cross-validation was used to assess model prediction accuracy (Fig. 3b). Here, the confirmatory dataset was randomly divided into ten equal subsamples. One subsample was used as the independent validation data (10% of the confirmatory dataset) and was always excluded from training; the remaining nine subsamples were used as training data (90% of the confirmatory dataset). The training data were
further divided randomly into a training set (70% of the training data), validation set (15% of the training data), and testing set (15% of the training data) during the training process. The training set was used for computing the gradient and updating the network weights and biases, the validation set was used to monitor the error and control model stop, and the testing set was used to monitor whether the model was over-fitted (Palacz et al., 2013). We compared performances in predicting the independent validation data from the ten-fold cross-validation and selected the optimal model based on the lowest root mean
square error (RMSE). Then we applied the optimal model to the exploratory dataset (Fig. 2) and evaluated model performance by calculating error statistics. In our study, calculations were done in the MathWorks Matlab environment, using the Deep Learning Toolbox.

First, we compared the performance of one hidden layer vs. two hidden layers in predicting independent validation data. The number of neurons varied from $2^2$ to $2^8$ for the first hidden layer and was fixed at four in the second hidden layer for the two
hidden layers model; the number of neurons in the first layer was the same in the one hidden layer vs. two hidden layers model (Fig. 4). The ten-fold cross-validation showed that the model with two hidden layers performed better as the number of neurons increased. Second, in order to choose suitable training techniques and activation functions of the ANN model with two hidden layers, we tested three training functions (Gradient descent backpropagation (trainGD), Levenberg-Marquardt backpropagation (trainLM), and Scaled conjugate gradient backpropagation (trainSCG)), which differed in how the weights
are modified, and three transfer functions (Log-sigmoid transfer function (logsig), Hyperbolic tangent sigmoid transfer function (tansig), and Positive linear transfer function (poslin)) (Fig. 5). The output values of logsig, tansig and poslin were compressed onto [0, 1], [-1, 1], and [0, +∞], respectively (Fig. S1). As the number of neurons increased, the performances of trainGD and tansig became poor. Although there was no obvious difference between trainLM and trainSCG, the training technique trainSCG was selected and the transfer function logsig was applied to two hidden layers considering the overall
performance (Fig. 5). Third, in the training phase of the ANN model, the number of neurons was tested, varying from 4 to 128 for two hidden layers (Table S1). Best performance for both training data and independent validation data was obtained with 40 neurons in the first hidden layer and 16 neurons in the second layer. Finally, different combinations of input variables were tested to choose the optimal architecture of the ANN model (Table 2); best performance was obtained using longitude, latitude, month, T, S, DO, N, P and Si as input variables. The utility of these variables for predicting pH has a strong a priori basis: the
carbonate system thermodynamic relationships depend on both T and S (Lueker et al., 2000); a positive correlation is expected between DO and pH (Wootton et al., 2012) because of the role of photosynthesis and respiration in removing or generating $CO_2$ in the water; various nutrients influence phytoplankton growth and abundance, thereby increasing organic carbon fixation/uptake and increasing pH (Wootton et al., 2008, 2012). We found geographical information to be a powerful addition in improving the skill of the method (Table 2), allowing the network to learn spatio-temporal patterns that could not be
explained by other input variables (Sasse et al., 2013).

In order to avoid bias towards high-value inputs/outputs and to eliminate the dimensional influence of the data, all data used by the ANN model were normalized using the following equation (e.g., Sauzède et al., 2015, 2016):

$$x_{i,j} = \frac{2}{3} * \frac{x_{i,j} - mean(x_{i,j})}{\sigma(x_{i,j})} \qquad (1)$$

with σ the standard deviation of the considered input variables or output variable $pH_T$. Similar to the approach of Sauzède et

al. (2015, 2016), the longitude and month input variables were transformed as follows to account for the periodicity:

$$\text{slongitude} = \sin\left(\frac{Lon*\pi}{180}\right), \ \text{clongitude} = \cos\left(\frac{Lon*\pi}{180}\right) \ (2)$$

$$\text{smonth} = \sin\left(\frac{month*\pi}{6}\right), \ \text{cmonth} = \cos\left(\frac{month*\pi}{6}\right) \ (3)$$

The latitude variable was transformed into the range of the sigmoid function by dividing by 90, then normalized using (1).

## 3 Result and discussion

### 3.1 ANN model performance

To evaluate the performance of the ANN model, we compared model simulated $pH_T$ ($pH_T^M$) with corresponding observations ($pH_T^O$) using several statistical indices, including the mean absolute error (MAE), the coefficient of determination ($R^2$), and the root mean squared error (RMSE). The model simulated $pH_T$ with a RMSE of 0.04 and $R^2$ of 0.88 for the training data (90% of confirmatory dataset, Fig. 6a), and predicted $pH_T$ with a RMSE of 0.03 and $R^2$ of 0.93 for the independent validation data (10% of confirmatory dataset, Fig. 6b). The histogram of residuals in confirmatory dataset (Fig. 6c) showed that 68% of the residuals were within the RMSE of 0.04. In order to further explore where the ANN model may lead to large errors, we plotted distributions of differences ($pH_T^M$ - $pH_T^O$) with respect to the longitude and latitude (Fig. 7). The points with large errors are mainly concentrated in the longitude range [122.5°E, 123°E] and the latitude range [31°N, 32.5°N], in an area strongly influenced by the Changjiang Dilute Water (CDW). The reduced performance of the ANN model here may be primarily due to the strong seasonal oscillations of the Changjiang discharge (Dai and Trenberth, 2002). As a reference, the performance of some other empirical approaches, including MLR, multi-variate nonlinear regression (MNR), decision tree, random forest, and Support Vector Machine (SVM) regression, is shown in Table 3. The selected ANN model (Table 2, Model#10) showed better performance than the other tested approaches using the same input variables (Table 3).

### 3.2 ANN model validation using the exploratory dataset

To further assess the ability of the ANN model to estimate $pH_T$ on the ECS shelf, we applied the ANN model to an exploratory dataset not used in ANN model development and sampled during March, July, and October 2018 (Fig. 2). Scatterplots of retrieved $pH_T$ vs observations (Fig. 8a) showed an RMSE of 0.04, $R^2$ of 0.80 and MAE of 0.03, which is consistent with the performance of the training data (Fig. 6a). Although the RMSE for $pH_T$ we obtained here was higher than obtained in some previous studies (e.g., Juranek et al., 2011; Williams et al., 2016; Sauzède et al., 2017), these latter studies considered open ocean regions, not coastal seas. For example, Juranek et al. (2011) developed empirical algorithms to estimate pH with RMSE of 0.018 for data between 30-500 m in the NE subarctic Pacific; Williams et al. (2016) also developed empirical algorithms to predict pH with RMSE of 0.01 in the Southern Ocean; Sauzède et al. (2017) developed a neural network method to estimate pH with RMSE of 0.02 in the global ocean. As a further comparison we applied the CANYON model developed by Sauzède et al. (2017) to our coastal exploratory dataset (Fig. 8b), and obtained an RMSE of 0.09 and MAE of 0.06. It is not surprising that the ANN model (developed here for the ECS shelf) outperforms the CANYON model (developed for the global ocean) for predicting $pH_T$ on the ECS shelf. The carbon chemistry parameters in this region are not only under the direct impact of Taiwan Warm Current and remote control of the Kuroshio water intrusion into the shelf, but are also significantly controlled by seasonal variations of the Changjiang discharge (e.g., Isobe and Matsuno, 2008; Chen et al., 2008; Chou et al., 2009). Taking into account the highly complex hydrographic, biological and chemical conditions, the accuracy of $pH_T$ presented is promising.

### 3.3 ANN model sensitivity to environmental input variables

To assess the ANN model sensitivity to different environmental input variables, we added 5% perturbation for each environmental variable separately. Statistically, with 5% T errors added, the ANN model showed slight overestimation in $pH_T$, with mean bias (MB) of 0.0059, RMSE of 0.0079, and $R^2$ of 0.9949 (Fig. 9a); with 5% DO errors added, the ANN model also showed slight $pH_T$ overestimation, with MB of 0.0050, RMSE of 0.0090, and $R^2$ of 0.9934 (Fig. 9c); with 5% S errors added, the ANN model showed overestimation in pHT, with MB of -0.0111, RMSE of 0.0162, and $R^2$ of 0.9789 (Fig. 9b). These results suggested that the ANN model responded to T and DO errors in a positive way, S errors in a negative way. The positive response to increasing DO reflects positive correlation between $pH_T$ and DO (Cai et al., 2011), which can be attributed to the processes of photosynthesis (generating DO and removing $CO_2$, hence increasing pH) and aerobic respiration (consuming DO and generating $CO_2$, hence lowering pH); the negative response to increasing S reflects the influence of the (lower salinity) Changjiang discharge, carrying large amounts of nutrients that fuel increased primary production (uptake of nutrients and $CO_2$, hence raising the pH) in surface waters during warm seasons (Gong et al., 2011). It was found that the ANN model was insensitive to nutrients errors (Fig. 9d-9f) and most sensitive to S errors (Fig. 9b), followed by DO and T errors.

### 3.4 ANN model application

#### 3.4.1 Comparison

In order to retrieve monthly $pH_T$ on the ECS shelf, the monthly T, S, DO, N, P and Si from the Changjiang Biology Finite-Volume Coastal Ocean Model (FVCOM) (http://47.101.49.44/wms/demo) were fed into the ANN model as input variables. The resolution of the Changjiang Biology FVCOM output is 1-10 km in the horizontal, 10 depth levels in the vertical, and day in the temporal (refered Ge et al., (2013) for detail information). Comparisons of monthly-average FVCOM model variables with surface and bottom observations on the ECS shelf showed that simulated T was close to observed values (Fig. S2a), simulated S was also close to observed values except at the bottom in August 2013 and at the surface in July 2016 (Fig. S2b), simulated DO was higher than observed at the bottom (Fig. S2c), and simulated nutrients were higher than observed at the surface (Fig. S2d-S2f). Comparisons of monthly average $pH_T$ from the FVCOM biogeochemical model with $pH_T$ retrieved by the ANN model suggested that the ANN model can potentially provide a more accurate $pH_T$ (Fig. S3). The possible reason was that the carbonate system from the Changjiang Biology FVCOM was not optimized due to challenges obtaining sufficient boundary information.

Considering the discreteness and discontinuity of the sampling sites, we compared $pH_T$ retrieved by the ANN model using the Changjiang Biology FVCOM output with the corresponding observations at some sites with repeated sampling for 3 to 4 years. These sites were A1-5 (123.0140°E, 32.2145°N), A1-6 (123.2750°E, 32.2679°N), A6-7 (122.9880°E, 30.7050°N), A6-9 (123.4990°E, 30.5723°N), A7-5 (123.4990°E, 30.2523°N), and A8-5 (123.4930°E, 29.9940°N). Overall, the retrieved $pH_T$ agrees well (within the ANN model accuracy: ANN±RMSE) with the observed values at the surface, except for three samples in summer (Fig. 10). There are relatively large deviations (greater than the RMSE of 0.04) in August 2013 at station A1-5 and A6-9, and in July 2016 at station A8-5. To illustrate the application performance in the water column, a scatterplot of retrieved $pH_T$ vs observations at six sites with repeated sampling for 3 to 4 years (Fig. 11) showed that the ANN model predicted $pH_T$ with a RMSE of 0.05 and $R^2$ of 0.71.

We further compared monthly $pH_T$ retrieved by the ANN model using the Changjiang Biology FVCOM output with in situ measured $pH_T$ values (Fig. 12). The agreement is good (within the ANN model accuracy: ANN±RMSE) here in winter, but large deviations (greater than the RMSE of 0.04) appear in summer. The reduced performance in summer can be attributed in large part a reduced performance of the Changjiang Biology FVCOM in predicting summertime input variables S, DO, and nutrients (Fig. S2).

### 3.4.2 Spatial and temporal patterns of ANN-derived pH$_T$

The temporal and spatial variations of monthly surface pH$_T$ from 2000-2016 based on Changjiang Biology FVCOM output are shown in Figure 13. During the dry season (November to March of the next year), pH$_T$ values vary from ~7.62 to ~8.24. Relatively higher pH$_T$ values are found in the southeastern of the study area (Chou et al., 2011), whereas lower pH$_T$ values are found in the northeastern of the study area. During the wet season (April to October), pH$_T$ values vary from ~7.77 to ~8.35, water of higher pH$_T$ corresponded well to the seasonal dispersion of the Changjiang Dilute Water (Chou et al., 2009, 2013). Water of higher pH$_T$ is found in the center of the study area during April, spreads to the southwestern part of the study area (along the coast of China) during May and June, shifts to the northeastern part of the study area during August. In September and October, water of higher pH$_T$ is found in the southeastern part of the study area, strongly influenced by the Taiwan Warm Current (Qu et al., 2015).

A clear seasonality is that surface pH$_T$ gradually increases during spring (March to May), after which it gradually decreases during summer and fall (June to November) (Fig. 14). The surface pH$_T$ displays its maximum in May and minimum in December, and the pH$_T$ varies seasonally by up to ~0.3 unit. Larger changes in pH were also discovered in the Washington Shelf, the pH varied ~1.0 unit over the seasons and ~1.5 unit spanning 8 years (Wootton et al., 2008). Accordingly, seasonal dynamics of surface pH$_T$ can be mainly attributed to temperature changes and strong biological activities (production and respiration processes) over the season. From March to June, a rapid increase in surface pH$_T$ indicates that production increases faster than respiration, which can be reflected in the drop in surface phosphate (Fig. S5d) and apparent oxygen utilization (AOU) (Fig. S5c). It may be driven by the Changjiang discharge (Fig. S4), which carries large amount of nutrients, result in stronger primary production in warm seasons under the combined action of nutrients and suitable temperature (Gong et al., 2011). From July to October, although surface temperature remains at a high level (Fig. S5a), the rise in surface AOU (Fig. S5c) suggest a decrease in primary production or increase of respiration, which leads to a gradual drop in surface pH$_T$ (Wootton et al., 2012). It implies respiration processes dominate relative to primary production during summer and fall.

### 4 Summary and conclusions

We have developed an artificial neural network (ANN) model, demonstrated its reliability, and used it to retrieve monthly pH$_T$ for the period 2000-2016 on the East China Sea shelf. We trained this ANN model using 11 cruise datasets from 2013 to 2017. In order to choose the optimal architecture of the ANN model, we tested different training and transfer functions, the number of neurons in two hidden layers, and different combinations of input variables. We also validated the reliability of the ANN model with a root mean square error accuracy of 0.04 using three cruises in 2018 as exploratory dataset. The ANN model responded to temperature and dissolved oxygen errors in a positive way, salinity errors in a negative way, and was most sensitive to salinity errors, followed by dissolved oxygen and temperature errors. We also retrieved monthly-average pH$_T$ using the ANN model in combination with input variables from the Changjiang Biology Finite-Volume Coastal Ocean Model (FVCOM).

The approach has several potential applications. First, it can provide estimates of seawater pH$_T$ with known accuracies for the East China Sea shelf and the period 2013-2018. Within this region the model could be used as a cost-effective way to handle restrictions of marine observations conducted from ships, such as coarse resolution and under-sampling of carbonate system variables. Second, while the ANN model is not a replacement for direct measurements of the carbonate system, it may be a valuable tool for understanding the seasonal variation of pH$_T$ in poorly observed regions. Third, this approach can be applied to other regions to predict pH by suitably adapting the input variables and network structure using local dataset. The MATLAB code used in this study to develop and apply the ANN model is freely available, and is accompanied by a README file providing detailed guidance on how to use and adapt the code.

**Code and data availability**

Matlab code of the ANN model for $pH_T$ estimation and datasets are available:

http://doi.org/10.5281/zenodo.3519219

The monthly-average input variables (T, S, DO, N, P, Si) from the Changjiang Biology Finite-Volume Coastal Ocean Model and retrieved $pH_T$ values from 2000 to 2016 on the East China Sea shelf and three cruises data during 2018 used to evaluate the ANN model are available:

http://doi.org/10.5281/zenodo.3519236

Requests to access the raw data should be directed to Richard Bellerby: Richard.Bellerby@niva.no

Six stations with repeated sampling for 3 to 4 years and corresponding retrieved pH values from the Changjiang Biology FVCOM output are available: http://doi.org/10.5281/zenodo.3491747

**Video supplement**

Monthly distribution of surface $pH_T$ on the East China Sea shelf from 2000 to 2016 year:

http://doi.org/10.5281/zenodo.2672943

Profile distribution of $pH_T$ at 31°N on the East China Sea shelf from 2000 to 2016 year:

http://doi.org/10.5281/zenodo.2672929

**Author contribution**

Li, X. S. and Bellerby, R. contributed to the development of methodology and the design of the model. Ge, J. Z. provided ten cruises dataset from 2013 to 2017 year and the input variables from the Changjiang Biology Finite-Volume Coastal Ocean

Model Data. Liu, J. and Yang, A. Q. provided four cruises dataset from 2017 to 2018 year. Li, X. S. developed the manuscript with contributions from all co-authors.

**Acknowledgements**

This study was financially supported by the National Thousand Talents Program for Foreign Experts (grants No. WQ20133100150), Vulnerabilities and Opportunities of the Coastal Ocean (grants No. SKLEC-2016RCDW01), Marginal

Seas (MARSEAS) (grants SKLEC-Taskteam project), and Innovative Talents International Cooperation Training Project (grants No. China Scholarship Council-201913045). Richard Bellerby and Philip Wallhead were also supported by funding from the FRAM High North Research Centre for Climate and the Environment under the Ocean Acidification Flagship and the NIVA Land-Ocean Interactions Strategic Institute program. We deeply thank the people who worked on the cruises and in the laboratory.

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

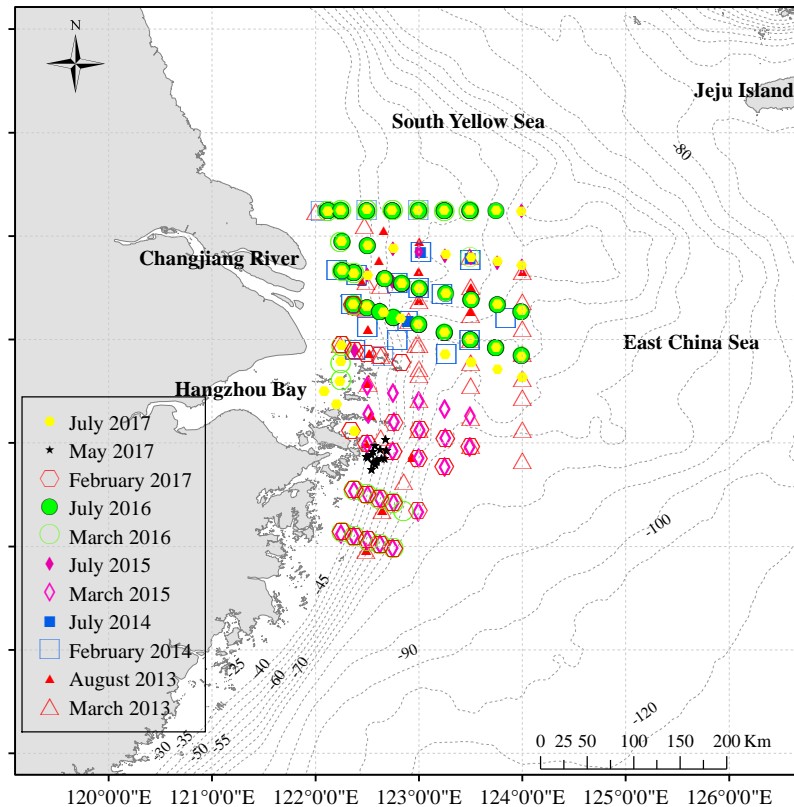

Figure 1: Sampling stations during 11 cruises (the confirmatory dataset) from 2013 to 2017 on the East China Sea shelf.

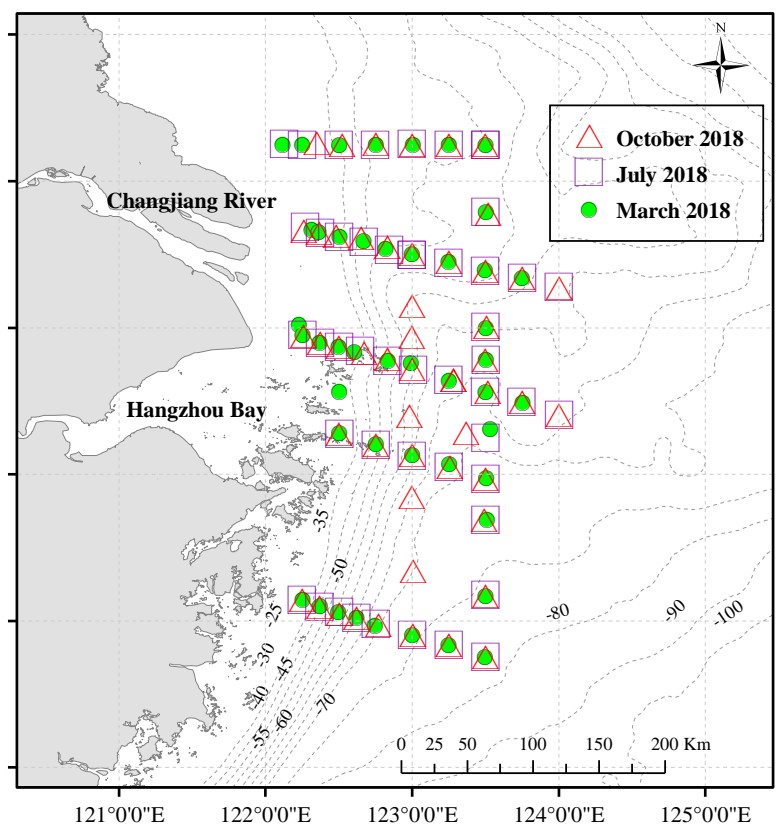

Figure 2: Sampling stations for three cruises (the exploratory dataset) used to extend the utility of the ANN model. The green circles represent March 2018, the purple squares represent July 2018, the red triangles represent October 2018.

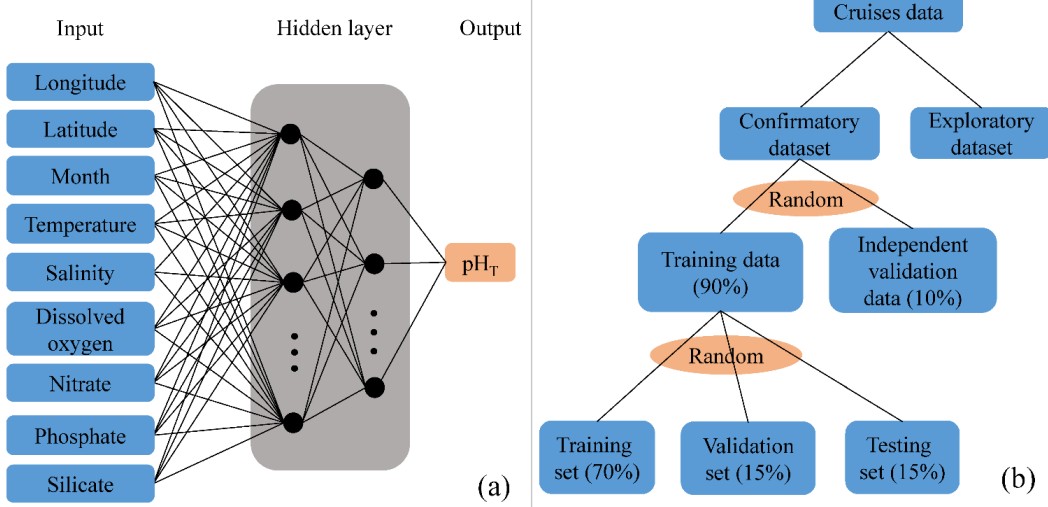

**Figure 3: Schematic representation of the neural network algorithm to retrieve pH$_T$. (a)-the architecture of the ANN model. Input variables are observed temperature, salinity, dissolved oxygen, nitrate, phosphate, and silicate together with the geolocation (longitude and latitude) and time (month) of sampling; (b)-data distribution diagram for training and prediction.**

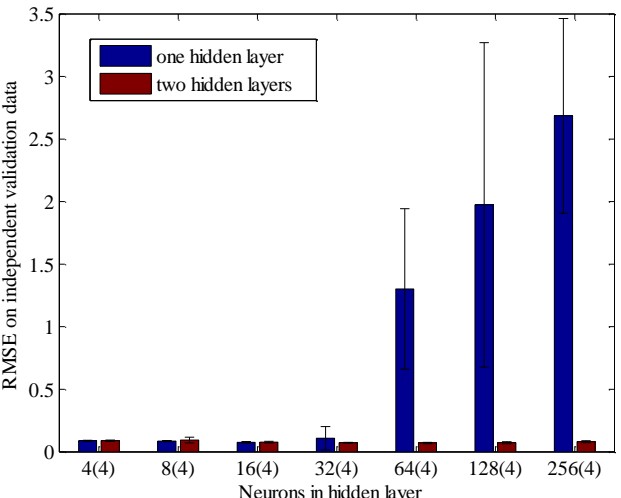

**Figure 4: Comparison of the performance of one hidden layer vs. two hidden layers in predicting independent validation data. The number of neurons in the first hidden layer was the same in the one hidden layer vs. two hidden layers model, numbers in parentheses show the number of neurons in the second hidden layer (for the two hidden layers model). Bars show the mean and standard deviation of the Root-Mean-Square-Error over a ten-fold cross-validation, for different numbers of neurons in the first hidden layer.**

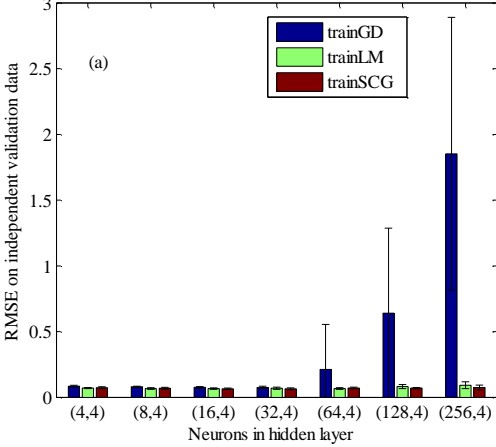 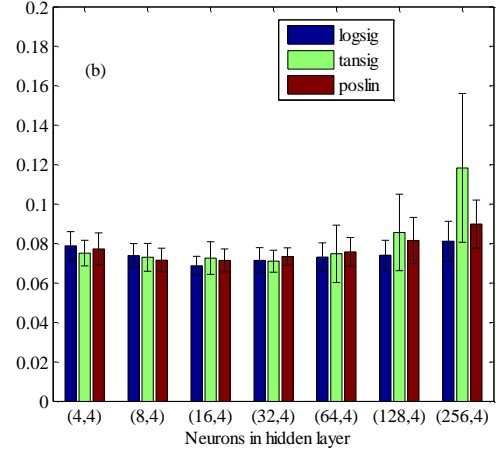

**Figure 5: Comparison of the performance of different training functions and transfer functions on independent validation data. (a)-three training functions: Gradient descent backpropagation (trainGD), Levenberg-Marquardt backpropagation (trainLM), and Scaled conjugate gradient backpropagation (trainSCG); (b) three transfer functions: Log-sigmoid transfer function (logsig), Hyperbolic tangent sigmoid transfer function (tansig), and Positive linear transfer function (poslin). Bars show the mean and**

 **standard deviation of the Root-Mean-Square-Error over a ten-fold cross-validation, for different numbers of neurons in the first hidden layer.**

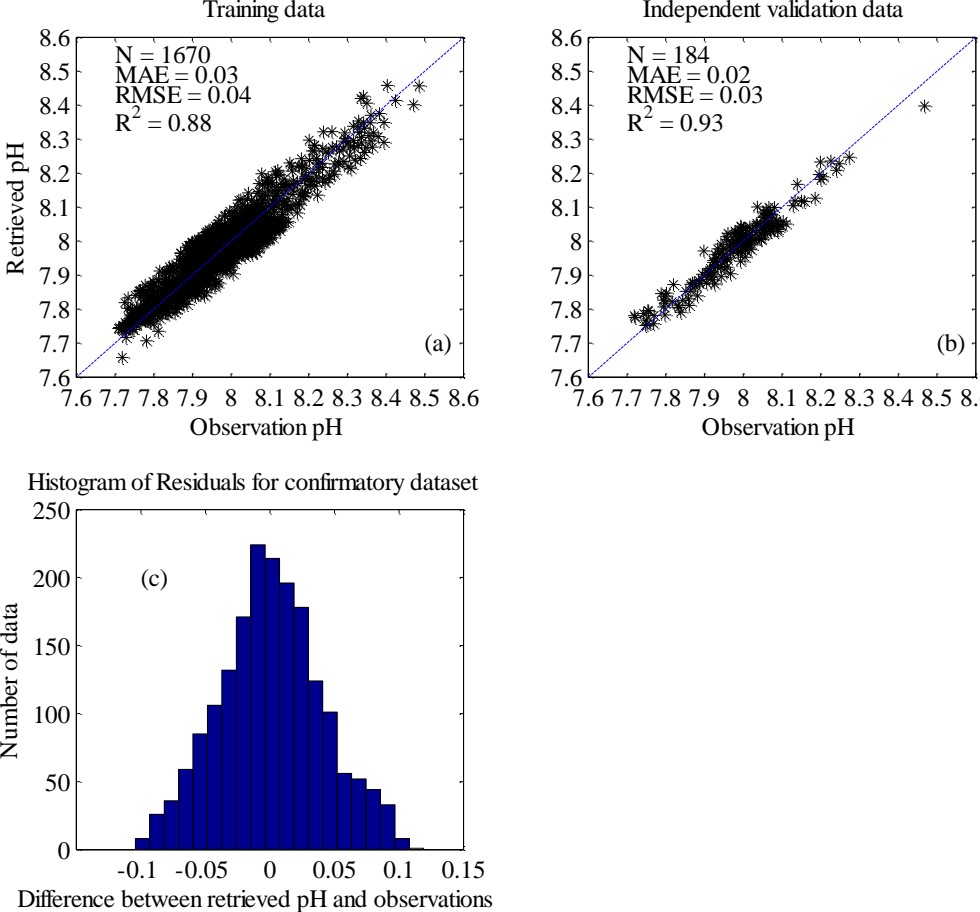

Figure 6: Comparison of pH_T retrieved by the ANN model with corresponding observations. (a)-Training data (90% of confirmatory dataset); (b)-Independent validation data (10% of confirmatory dataset); (c)-Histogram of residuals for confirmatory dataset. The 1:1 line is shown in each plot as visual reference. Three statistics are the mean absolute error (MAE), the coefficient of determination ($R^2$), and the root mean squared error (RMSE). N represents the number of data points.

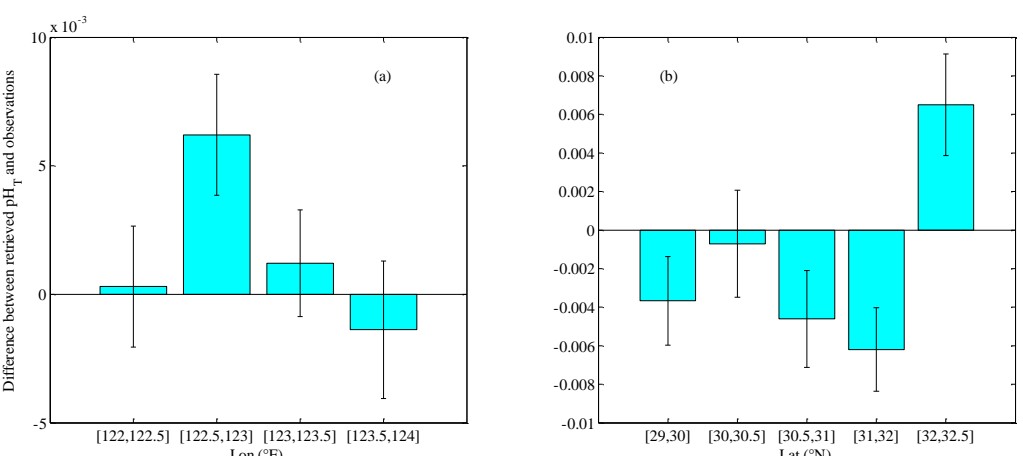

**Figure 7: Box plots of the differences between retrieved pH_T minus the observations. (a)-the differences vs longitude (Mean±SE); (b)-the differences vs latitude (Mean±SE). The height of each box represents the mean value of the differences, the whisker represents the standard error (SE) value of the differences.**

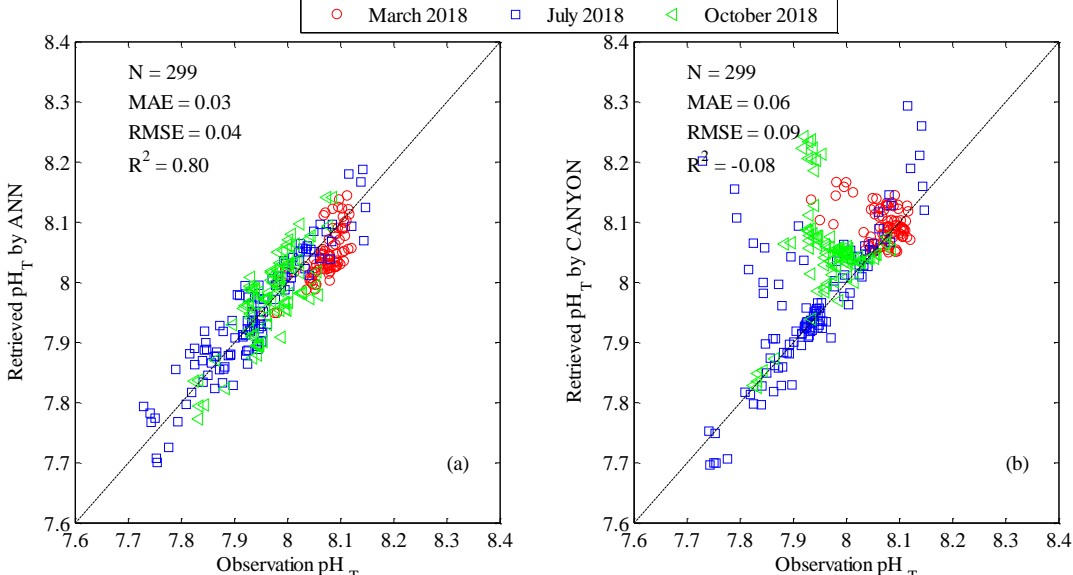

**Figure 8: Comparison of retrieved pH$_T$ with corresponding observations for exploratory dataset. (a)-pH$_T$ retrieved by the ANN model vs observations; (b)-pH$_T$ retrieved by CANYON (Sauzède et al., 2017) vs observations. The red circles represent March 2018, the blue squares represent July 2018, the green triangles represent October 2018. The 1:1 line is shown in the plot as visual reference.**

**Three statistics approaches used are the mean absolute error (MAE), the root mean squared error (RMSE), and the coefficient of determination (R$^2$). N represents the number of data points.**

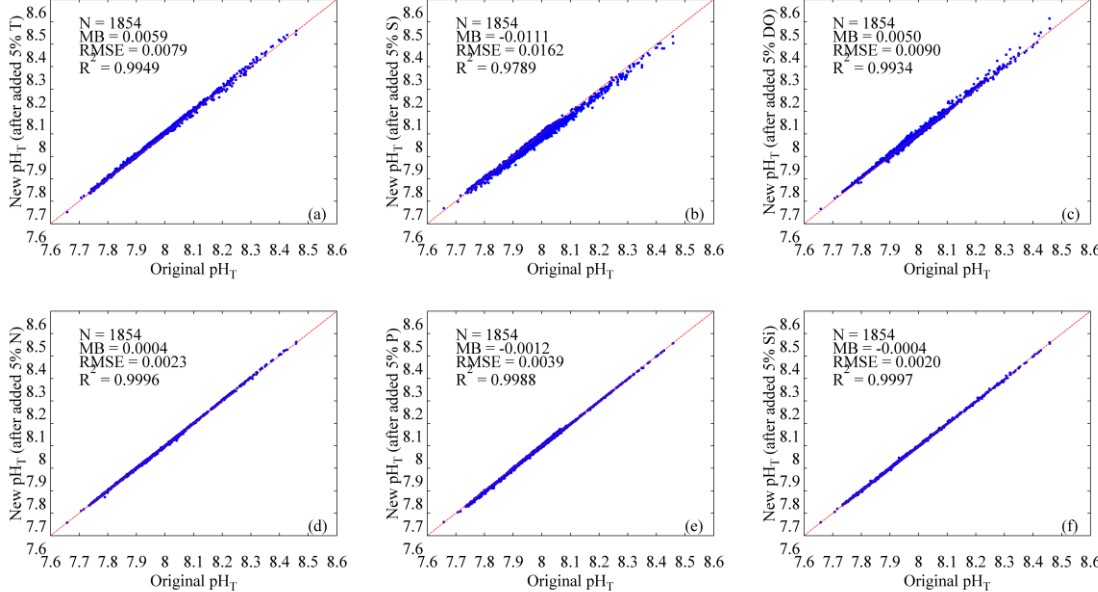

**Figure 9: Sensitivity of the ANN model for environmental input variables. (a)-temperature (T); (b) salinity (S); (c)-dissolved oxygen (DO); (d)-nitrate (N); (e)-phosphate (P); (f)-silicate (Si). Three statistics approaches used are the mean bias (MB), the root mean squared error (RMSE), and the coefficient of determination (R$^2$). N represents the number of data points.**


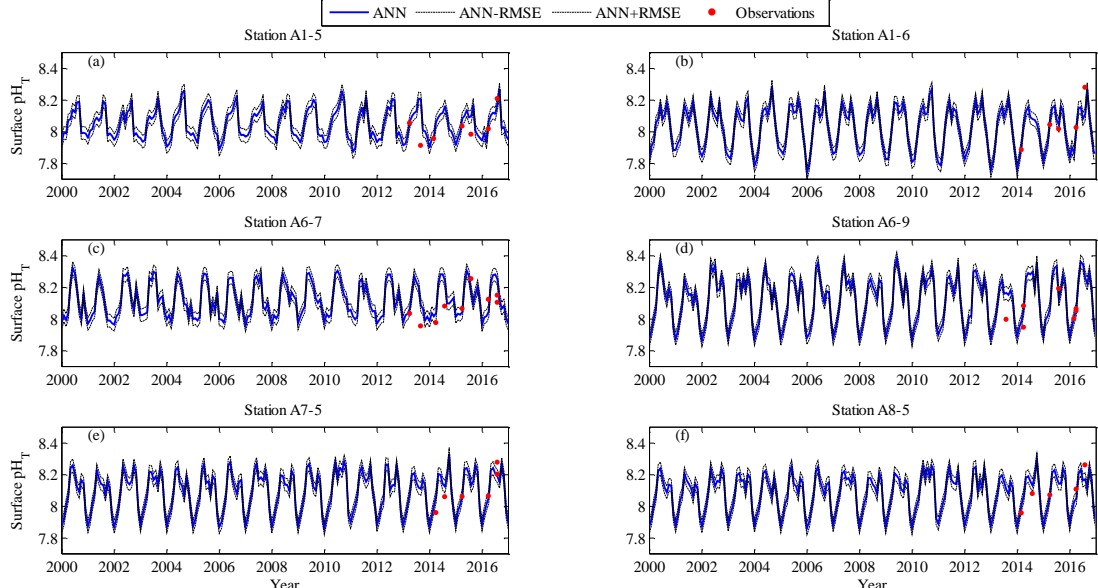

**Figure 10: Comparison of surface pH$_T$ retrieved by the ANN model using Changjiang Biology FVCOM output with corresponding observations at six sites repeated sampling for 3 to 4 years. Red dots represent observations pH$_T$, blue solid line represents retrieved pH$_T$, black dotted lines represent upper and lower bounds of the ANN model accuracy (ANN ± RMSE). (a)-station A1-5; (b)-station A1-6; (c)-station A6-7; (d)-station A6-9; (e)-station A7-5; (f)-station A8-5.**

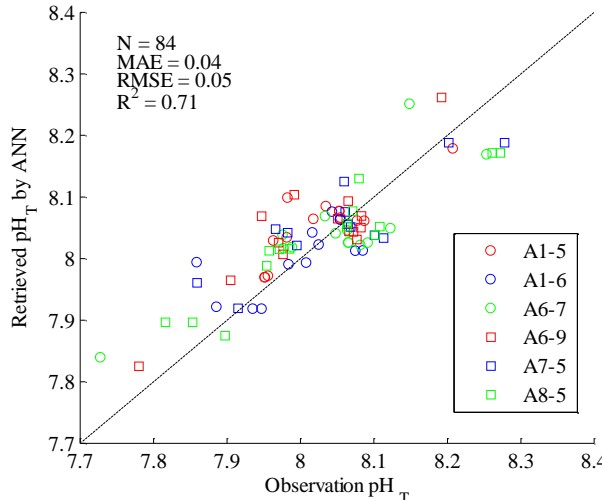

**Figure 11: Comparison of water column pH$_T$ retrieved by the ANN model using Changjiang Biology FVCOM output with corresponding observations at six sites repeated sampling for 3 to 4 years. The 1:1 line is shown in the plot as a visual reference. Skill statistics include the mean absolute error (MAE), the coefficient of determination (R$^2$), and the root mean squared error (RMSE). N represents the number of data points.**

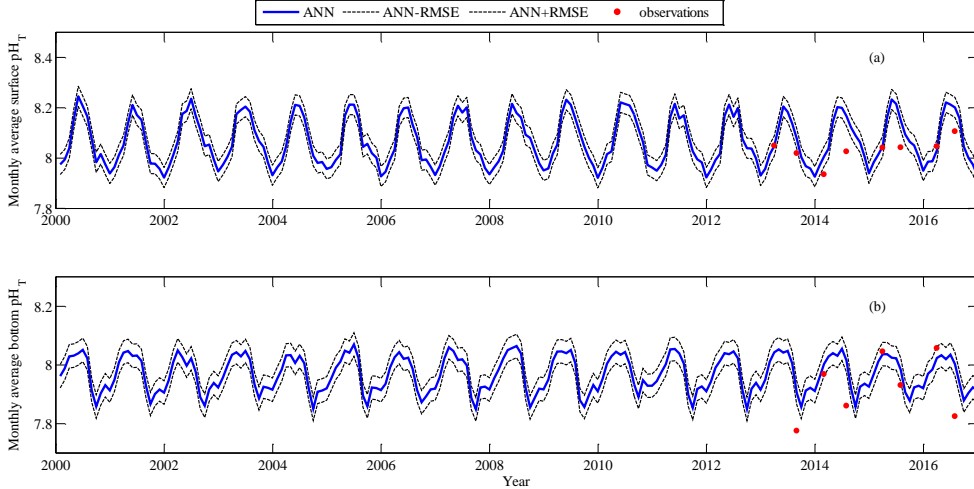

**Figure 12: Comparison of monthly average pH$_T$ on the East China Sea shelf. Blue solid line represents retrieved pH$_T$ by the ANN model using Changjiang Biology FVCOM output; black dotted lines represent upper and lower bounds of the ANN model accuracy (ANN ± RMSE); red points show monthly-average pH$_T$ observations from 2013 to 2016. (a)-surface; (b)-bottom.**

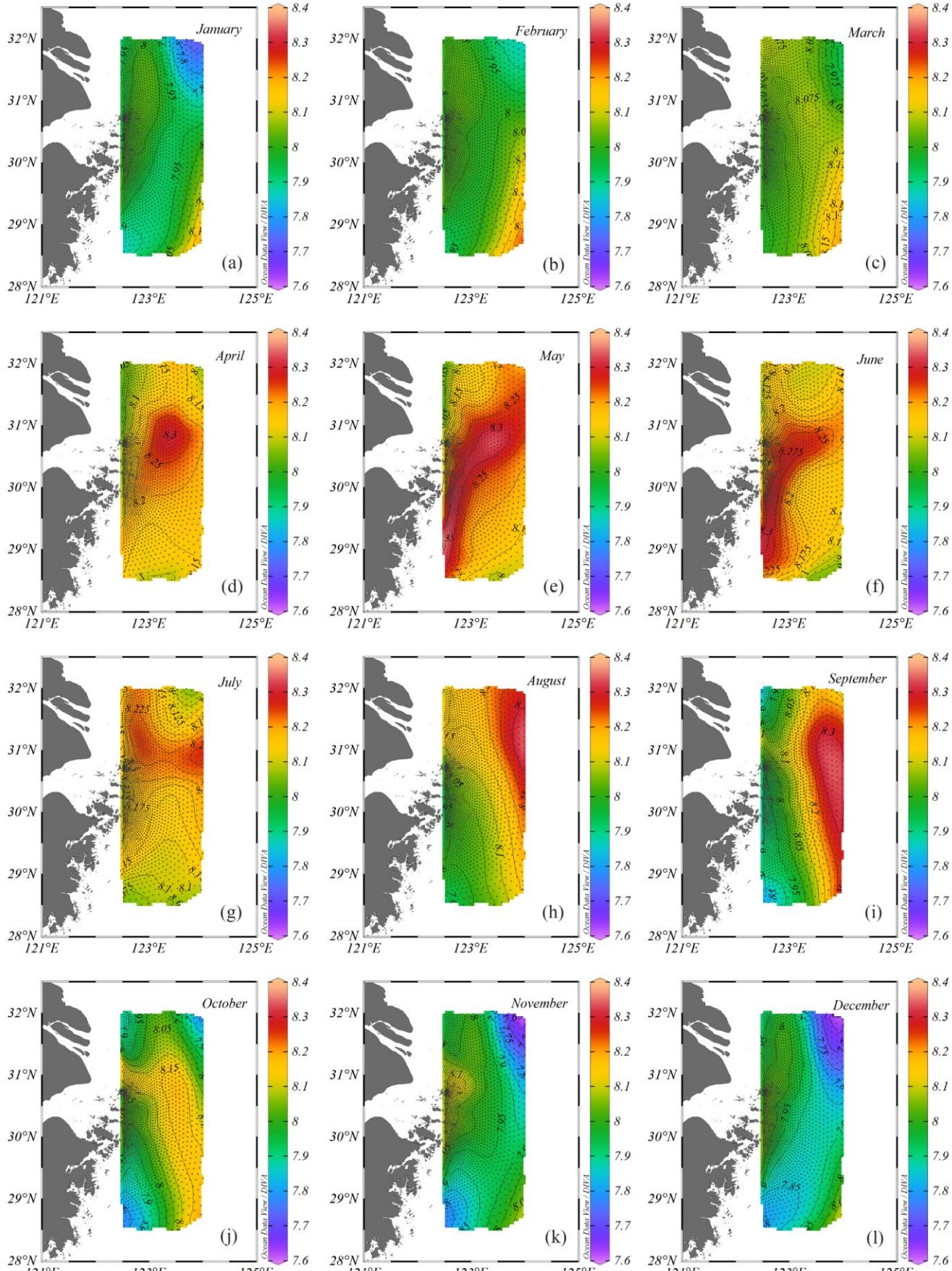

**Figure 13: Spatial distribution of monthly average surface pH$_T$ retrieved by the ANN model using Changjiang Biology FVCOM output. (a)-January; (b)-February; (c)-March; (d)-April; (e)-May; (f)-June; (g)-July; (h)-August; (i)-September; (j)-October; (k)-November; (l)-December.**


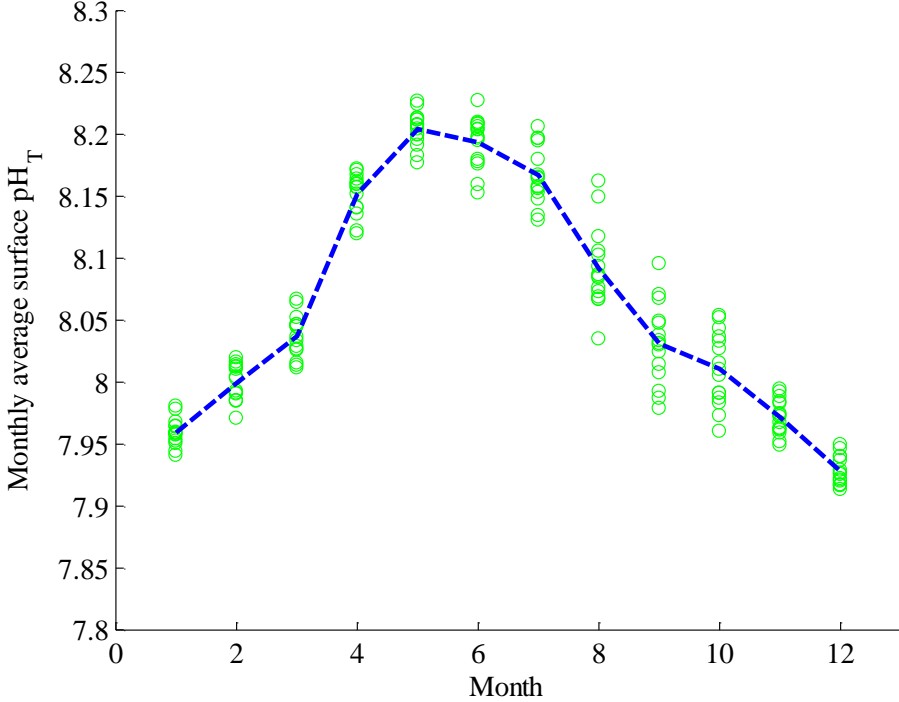

**Figure 14: Seasonal cycles of surface pH$_T$ on the East China Sea shelf from 2000-2016. The green circles represent monthly regional average, the blue dashed represents mean value of each month.**


**Table 1: Field survey information and measurements of water temperature, salinity, dissolved oxygen, nitrate, phosphate, silicate and pH$_T$ (Mean±SE).**

| Sampling period | Temperature (℃) | Salinity | Dissolved oxygen (mmol m$^{-3}$) | Nitrate (mmol m$^{-3}$) | Phosphate (mmol m$^{-3}$) | Silicate (mmol m$^{-3}$) | pH$_T$ |
|---|---|---|---|---|---|---|---|
| March 4$^{th}$-20$^{th}$, 2013 | 11.54±1.34 | 32.04±2.26 | 275.28±19.30 | 12.25±8.25 | 0.58±0.17 | 17.54±7.65 | 8.19±0.04 |
| August 17$^{th}$-28$^{th}$, 2013 | 23.45±3.17 | 32.32±2.91 | 142.22±63.45 | 12.16±8.05 | 0.55±0.32 | 16.47±12.18 | 8.04±0.18 |
| February 21$^{th}$-28$^{th}$, 2014 | 9.56±2.38 | 32.14±1.78 | 293.07±19.52 | 11.92±9.17 | 0.59±0.18 | 12.52±6.50 | 8.10±0.04 |
| July 10$^{th}$-17$^{th}$, 2014 | 21.66±2.13 | 29.50±5.10 | 186.44±43.29 | 21.57±22.10 | 0.57±0.46 | 21.45±17.76 | 8.07±0.11 |
| March 11$^{th}$-21$^{th}$, 2015 | 11.42±1.44 | 31.57±2.60 | 279.72±15.29 | 22.04±18.88 | 0.81±0.35 | 16.48±11.64 | 8.19±0.03 |
| July 9$^{th}$-20$^{th}$, 2015 | 22.14±1.55 | 29.73±4.71 | 207.32±56.12 | 19.73±18.62 | 0.60±0.42 | 20.87±17.48 | 8.13±0.09 |
| March 7$^{th}$-19$^{th}$, 2016 | 10.77±2.02 | 30.85±2.92 | 284.00±31.40 | 20.26±12.80 | 0.82±0.25 | 19.17±11.62 | 8.20±0.05 |
| July 4$^{th}$-28$^{th}$, 2016 | 23.19±3.19 | 28.17±6.67 | 122.90±49.97 | 25.77±23.60 | 0.63±0.46 | 28.56±25.03 | 8.06±0.16 |
| February 15$^{th}$-28$^{th}$, 2017 | 11.03±2.57 | 32.00±2.43 | 296.21±21.27 | 12.30±9.13 | 0.56±0.18 | 13.09±7.45 | 8.13±0.05 |
| May 12$^{th}$-24$^{th}$, 2017 | 17.71±1.54 | 29.62±2.79 | 171.58±49.52 | 12.60±4.83 | 0.29±0.24 | 10.95±4.29 | 8.08±0.13 |
| July 20$^{th}$-30$^{th}$, 2017 | 24.85±3.41 | 27.70±6.31 | 192.11±76.55 | 20.57±23.23 | 0.42±0.34 | 19.28±18.92 | 8.09±0.18 |


**Table 2: Different model structures and their performance in the training step. The variables (Lon (longitude), Lat (latitude), Month (month), T (temperature), S (salinity), DO (dissolved oxygen), N (nitrate), P (phosphate), Si (silicate)) marked with 1 represent the input variables. Skill statistics include the coefficient of determination (R$^2$), the root mean squared error (RMSE), and the mean**

**absolute error (MAE).**

| Model | Lon | Lat | Month | T | S | DO | N | P | Si | Training data | | | Independent validation data | | |
|---|---|---|---|---|---|---|---|---|---|---|---|---|---|---|---|
| | | | | | | | | | | R$^2$ | RMSE | MAE | R$^2$ | RMSE | MAE |
| 1 | | | 1 | | | | | | | 0.40 | 0.092 | 0.068 | 0.47 | 0.076 | 0.058 |
| 2 | | | 1 | | | 1 | | | | 0.62 | 0.073 | 0.053 | 0.62 | 0.067 | 0.051 |

| | | | | | | | | | | | | | | | |
|---|---|---|---|---|---|---|---|---|---|---|---|---|---|---|---|
| 3 | | | | 1 | 1 | 1 | | | | 0.69 | 0.065 | 0.048 | 0.72 | 0.060 | 0.044 |
| 4 | | | | 1 | 1 | 1 | 1 | | | 0.76 | 0.057 | 0.044 | 0.77 | 0.052 | 0.041 |
| 5 | | | | 1 | 1 | 1 | | 1 | | 0.81 | 0.051 | 0.040 | 0.79 | 0.051 | 0.040 |
| 6 | | | | 1 | 1 | 1 | | | 1 | 0.77 | 0.056 | 0.044 | 0.79 | 0.054 | 0.043 |
| 7 | | | | 1 | 1 | 1 | 1 | 1 | | 0.80 | 0.053 | 0.042 | 0.79 | 0.051 | 0.041 |
| 8 | | | | 1 | 1 | 1 | | 1 | 1 | 0.81 | 0.051 | 0.040 | 0.81 | 0.049 | 0.039 |
| 9 | | | | 1 | 1 | 1 | 1 | | 1 | 0.76 | 0.058 | 0.044 | 0.77 | 0.054 | 0.044 |
| 10 | | | | 1 | 1 | 1 | 1 | 1 | 1 | 0.83 | 0.048 | 0.037 | 0.86 | 0.046 | 0.037 |
| 11 | | | 1 | 1 | 1 | 1 | 1 | 1 | | 0.85 | 0.046 | 0.035 | 0.87 | 0.043 | 0.032 |
| 12 | | | 1 | 1 | 1 | 1 | | 1 | 1 | 0.85 | 0.046 | 0.034 | 0.85 | 0.045 | 0.035 |
| 13 | | | 1 | 1 | 1 | 1 | 1 | | 1 | 0.82 | 0.049 | 0.036 | 0.84 | 0.050 | 0.036 |
| 14 | | | 1 | 1 | 1 | 1 | 1 | 1 | 1 | 0.84 | 0.046 | 0.035 | 0.87 | 0.045 | 0.033 |
| 15 | 1 | 1 | 1 | 1 | 1 | 1 | 1 | | | 0.86 | 0.044 | 0.033 | 0.79 | 0.046 | 0.034 |
| 16 | 1 | 1 | 1 | 1 | 1 | 1 | | 1 | | 0.87 | 0.043 | 0.032 | 0.87 | 0.044 | 0.034 |
| 17 | 1 | 1 | 1 | 1 | 1 | 1 | | | 1 | 0.87 | 0.043 | 0.033 | 0.82 | 0.045 | 0.035 |
| 18 | 1 | 1 | 1 | 1 | 1 | 1 | 1 | 1 | | 0.88 | 0.040 | 0.031 | 0.88 | 0.039 | 0.031 |
| 19 | 1 | 1 | 1 | 1 | 1 | 1 | | 1 | 1 | 0.87 | 0.042 | 0.032 | 0.87 | 0.042 | 0.033 |
| 20 | 1 | 1 | 1 | 1 | 1 | 1 | 1 | | 1 | 0.84 | 0.046 | 0.035 | 0.85 | 0.047 | 0.036 |
| 21 | 1 | 1 | 1 | 1 | 1 | 1 | 1 | 1 | 1 | 0.88 | 0.040 | 0.031 | 0.93 | 0.033 | 0.024 |

**Table 3: Model comparison between traditional empirical methods (MLR and MNR) and mechine-learning based empirical methods (Decision tree, Random Forest, and SVM). The statistics was derived from confimatory dataset (training data independent validation data) using input variables: T, S, DO, N, P, and Si. Note $R^2$ statistics in our study was based on the calculation of coefficient of determination, therefore negative $R^2$ could be derived if there were strong bias.**

| Model | Kernel Function | Input variables | RMSE | $R^2$ | MAE |
|---|---|---|---|---|---|
| MLR | - | T, S, DO, N, P, Si | 0.078 | 0.56 | 0.062 |
| MNR | - | T, S, DO, N, P, Si | 0.060 | 0.74 | 0.047 |
| Decision Tree | Simple Tree | T, S, DO, N, P, Si | 0.064 | 0.71 | 0.047 |
| | Medium Tree | T, S, DO, N, P, Si | 0.060 | 0.74 | 0.044 |
| | Complex Tree | T, S, DO, N, P, Si | 0.061 | 0.73 | 0.043 |
| Random Forest | Boosted Trees | T, S, DO, N, P, Si | 0.340 | -7.51 | 0.339 |
| | Bagged Trees | T, S, DO, N, P, Si | 0.056 | 0.77 | 0.04 |
| SVM | Linear | T, S, DO, N, P, Si | 0.079 | 0.55 | 0.061 |
| | Quadratic | T, S, DO, N, P, Si | 0.061 | 0.73 | 0.046 |
| | Cubic | T, S, DO, N, P, Si | 0.060 | 0.74 | 0.043 |
| | Fine Gaussian | T, S, DO, N, P, Si | 0.064 | 0.70 | 0.042 |
| | Medium Gaussian | T, S, DO, N, P, Si | 0.054 | 0.79 | 0.041 |
| | Coarse Gaussian | T, S, DO, N, P, Si | 0.069 | 0.65 | 0.054 |
