# Peer review of "Retrieving monthly and interannual $pH_T$ on the East China Sea shelf using an artificial neural network: ANN- $pH_T$ -v1"

_Geoscientific Model Development, 2019_

## Referee Comment (RC1) · Richard Mills (Referee) · 5 Jun 2020

The authors describe the development of an artificial neural network model (ANN) for predicting water column total scale pH (pH_T) in the waters of the East China Sea shelf from observations of pH_T, temperature (T), salinity (S), dissolved oxygen (DO), nitrate (N), phosphate (P) and silicate (Si), together with latitude, longitude, and time of samples. The ANN is trained using data from eleven cruises conducted between 2013-2017, comprising 1854 records, and the model performance is assessed in three different ways: ten-fold cross validation, application to new data from cruises in 2018, and using prognostic variables from the Changjian Biology Finite-Volume Coastal Ocean

Model (FVCOM) as input to the ANN. The ANN appears to perform well in the authors' tests.

This paper is well-written, with excellent English and a logical development of ideas. In its current form, however, I think that the paper is a promising beginning but does not seem complete. I came away from my reading of the paper with the following major questions/concerns which, if addressed, will greatly improve the quality of the paper:

First, since the paper has been submitted to a model development journal, I would like to see more information on how and why the authors arrived at the particular form of the machine-learning model they used, and how this model performed against some other possible model architectures. The authors have used a feed-forward multilayer perceptron network with two hidden layers (with 40 neurons in the first layer and 16 in the second) and full connectivity between the layers. Why did the authors decide on two layers, and how did they choose the number of neurons in each layer? (They do state that they tried varying the number of neurons in each layer, but don't give further details.) How did they choose the activation function? And why did they choose a neural network, instead of another approach such as k-nearest neighbors, random forest regression, or support vector regression? When I first started working in machine learning, around two decades ago, it would not have been expected for authors to try a variety of different types of models, as this would likely involve substantial code development effort, as well as possibly significant computational expense for training models. Today, however, it is easy to try many different models, as code provided in many easily obtained packages such as Scikit-learn or those provided by Matlab (the environment that the authors use for this study), and it is becoming the norm for papers presenting the development of machine-learning models to compare several types to determine the one that performs best for the chosen task. I would like to see some comparison against other models (some of the ones easily constructed using Matlab) to demonstrate that the ANN is the most appropriate choice.

Second, the authors do a good job of citing other papers in which authors have used

similar ANN approaches for similar biogeochemical prediction tasks in marine waters, and compare the RMSE of their model with published values from other models. I think that the paper would be greatly improved if the authors could do a direct comparison. For instance, the authors cite the CANYON neural network model of Sauzede et al., 2017, which has been developed for the global ocean, but note that "coastal seas tend to show greater temporal and spatial variability than open oceans", which I believe is an argument for why they developed the model presented in their paper. I can easily imagine that the model presented here will outperform the CANYON model for prediction on the East China Sea shelf, but I think it would be interesting for the authors to demonstrate this: The CANYON model appears to be freely available online, and it would be interesting to see how much better a model trained speficially for the East China Sea shelf will outperform one developed for the global ocean.

Finally, the authors perform an intersting study in which they use prognostic variables from the Changjian Biology Finite-Volume Coastal Ocean Model (FVCOM) as input to their ANN model in order to recover the $pH_T$. I am not a marine biogeochemistry modeler, so perhaps I am missing something obvious, but I am guessing that mechanistic models like FVCOM can provide prognostic $pH_T$. Is this available from the FVCOM runs that were used, or could it be obtained using FVCOM, or ROMS, or another, similar model? If so, how would the prognostic $pH_T$ from FVCOM (or similar) compare to the $pH_T$ from the authors' own ANN model? And what is the motivation for using the ANN? Is it because it can potentially provide a more accurate $pH_T$, or because it can provide $pH_T$ for situations in which it is not desirable to run a forward simulation or reanalysis to get the $pH_T$, or some other reason? This may be obvious to an marine biogeochemist, but I and many of the readers of GMD don't have this expertise. The motivation needs to be explained for the general GMD audience.

Detailed comments:

Lines 34-35: The authors state, while comparing ANNs to multiple linear regression, that ANNs have the advantage of not requiring 'an a priori model but rather "learn" the

model from existing data'. I think it would be more precise to say that they are non-parametric models and do not require assuming any underlying statistical distribution.

Lines 75 and 78: The authors say that samples were "poisoned" by addition of HgCl2. I think it may be more idiomatic to say "sterilized".

Line 81: "The final number of data used by the ANN model was 1854". I would say the final number of "observations" or "records", to be precise.

Line 94: The authors talk about a model being "over-matched". I believe that "over-fitted" is the term they mean.

There are problems with low resolution for all of the figures. Figure 1 is not really even readable. Figures need to be re-generated with much higher resolution, or using vector, rather than raster, formats.

---

## Author Comment (AC1) · 17 Jun 2020

**Our response to referee comment in the interactive discussion**

Dear Referee, dear Editor,

We would like to thank you very much for your positive comment and constructive suggestion to our manuscript "*Retrieving monthly and interannual $pH_T$ on the East China Sea shelf using an artificial neural network: ANN-pH_T-v1*".

In this document, we would like to provide our response to the referee comment posted by Richard Mills and to outline the corresponding changes to the manuscript. We will represent the referee comment in **bold** font, and our response in normal font. Quotations from the original manuscript will be in *italics*, changes as part of the manuscript revision will be highlighted as underlined. For the sake of clarity and brevity, we have omitted the introductory parts of the referee report (this omittance is marked as [...]).

We hope that our response together with the revision of the manuscript sufficiently addresses the referee' concerns.

Sincerely,

Xiaoshuang Li (on behalf of the author team)

**Referee comment by Richard Mills**

**[…] I came away from my reading of the paper with the following major questions/concerns which, if addressed, will greatly improve the quality of the paper:**

1. **First, since the paper has been submitted to a model development journal, I would like to see more information on how and why the authors arrived at the particular form of the machine-learning model they used, and how this model performed against some other possible model architectures. The authors have used a feed-forward multilayer perceptron network with two hidden layers (with 40 neurons in the first layer and 16 in the second) and full connectivity between the layers. Why did the authors decide on two layers, and how did they choose the number of neurons in each layer? (They do state that they tried varying the number of neurons in each layer, but don't give further details.) How did they choose the activation function? And why did they choose a neural network, instead of another approach such as k-nearest neighbors, random forest regression, or support vector regression? When I first started working in machine learning, around two decades ago, it would not have been expected for authors to try a variety of different types of models, as this would likely involve substantial code development effort, as well as possibly significant computational expense for training models. Today, however, it is easy to try many different models, as code provided in many easily obtained packages such as Scikit-learn or those provided by Matlab (the environment that the authors use for this study), and it is becoming the norm for papers presenting the development of machine-learning models to compare several types to determine the one that performs best for the chosen task. I would like to see some comparison against other models (some of the ones easily constructed using Matlab) to demonstrate that the ANN is the most appropriate choice.**

We thank the referee for the suggestion: the required details should, in fact, be provided to the reader. We will add, in the revised manuscript, the corresponding information (Why did the authors decide on two layers, and how did they choose the number of neurons in each layer? How did they choose the activation function? And why did they choose a neural network, instead of another approach such as k-nearest neighbors, random forest regression, or support vector regression?)—as follows ll. 112-128 and 160-163 of the revised manuscript:

ll. 112-128 of the revised manuscript

*In our study, calculations were done in the MathWorks Matlab environment, using the Deep Learning Toolbox.*

*Firstly, we compared the performance of one hidden layer and two hidden layers on independent validation data, the number of neurons varied from $2^2$ to $2^8$ for one hidden layer and fixed at four in the second hidden layer for two hidden layers (Fig. 4), the result of ten-fold cross-validation showed that the model with two hidden layers performed better as the number of*

*neurons increased. Secondly, in order to choose suitable training techniques and activation functions of the ANN model with two hidden layers, three training functions (Gradient descent backpropagation (trainGD), Levenberg-Marquardt backpropagation (trainLM), Scaled conjugate gradient backpropagation (trainSCG)), which differ on how the weights are modified, and three transfer functions (Log-sigmoid transfer function (logsig), Hyperbolic tangent sigmoid transfer function (tansig); Positive linear transfer function (poslin)) were tested (Fig. 5). The output values of logsig, tansig and poslin are squashed into [0, 1], [-1, 1], and [0, +∞], respectively (Fig. S1). The result demonstrated that as the number of neurons increased, the performances of trainGD and tansig became poor. Although there was no obvious difference between trainLM and trainSCG, here training technique trainSCG was selected and transfer function logsig was applied to two hidden layers considering the overall performance (Fig. 5). Thirdly, in the training phase of the ANN model, the number of neurons was tested, varying from 4 to 128 for two hidden layers (Table S1). It was found that the number of neurons was set to 40 in the first hidden layer and 16 in the second layer, the ANN model showed the best performance for both training data and independent validation data. Finally, different combinations of input variables were tested to choose the optimal architecture of the ANN model (Table 2). The result showed that the performance of the ANN model was optimal when longitude, latitude, month, T, S, DO, N, P and Si were used as input variables.*

[Figure]

**Figure 4 (revised): Comparison of the performance of one hidden layer and two hidden layers on independent validation data. The result displayed are the mean and standard deviation of ten-fold cross-validation for each number of neurons in the hidden layer. The number in parentheses presents the number of neurons in the second hidden layer for two hidden layers.**

[Figure]

[Figure]

**Figure 5 (revised): Comparison of the performance of different training functions and transfer functions on independent validation data. (a)-three training functions: Gradient descent backpropagation (trainGD), Levenberg-Marquardt backpropagation (trainLM), Scaled conjugate gradient backpropagation (trainSCG); (b) three transfer functions: Log-sigmoid transfer function (logsig), Hyperbolic tangent sigmoid transfer function (tansig); Positive linear transfer function (poslin). The result displayed are the mean and standard deviation of ten-fold cross-validation for each number of neurons in the hidden layer.**

[Figure]

**Figure S1 (revised): Comparison of three transfer functions. (a)-Log-sigmoid transfer function (logsig); (b) Hyperbolic tangent sigmoid transfer function (tansig); (c)-Positive linear transfer function (poslin).**

**Table S1 (revised): The performance of different number of neurons for two hidden layers in the training step. Three statistics are the coefficient of determination ($R^2$), the root mean squared error (RMSE), and the mean absolute error (MAE).**

| Model | Number of neurons | | Training data | | | Independent validation data | | |
|---|---|---|---|---|---|---|---|---|
| | first hidden | second hidden | $R^2$ | RMSE | MAE | $R^2$ | RMSE | MAE |
| 1 | 4 | 4 | 0.68 | 0.071 | 0.054 | 0.67 | 0.072 | 0.057 |
| 2 | 8 | 4 | 0.70 | 0.070 | 0.050 | 0.67 | 0.069 | 0.050 |
| 3 | 16 | 4 | 0.76 | 0.062 | 0.045 | 0.76 | 0.062 | 0.045 |
| 4 | 32 | 4 | 0.74 | 0.063 | 0.046 | 0.79 | 0.062 | 0.048 |
| 5 | 40 | 4 | 0.76 | 0.062 | 0.044 | 0.76 | 0.061 | 0.045 |
| 6 | 64 | 4 | 0.79 | 0.058 | 0.041 | 0.78 | 0.056 | 0.043 |
| 7 | 128 | 4 | 0.76 | 0.062 | 0.045 | 0.74 | 0.062 | 0.044 |
| 8 | 8 | 8 | 0.73 | 0.065 | 0.047 | 0.73 | 0.065 | 0.048 |
| 9 | 16 | 8 | 0.78 | 0.059 | 0.042 | 0.78 | 0.058 | 0.044 |
| 10 | 32 | 8 | 0.78 | 0.059 | 0.042 | 0.83 | 0.053 | 0.039 |
| 11 | 40 | 8 | 0.79 | 0.059 | 0.042 | 0.77 | 0.055 | 0.040 |
| 12 | 64 | 8 | 0.77 | 0.061 | 0.044 | 0.76 | 0.059 | 0.042 |
| 13 | 128 | 8 | 0.77 | 0.060 | 0.042 | 0.79 | 0.059 | 0.043 |
| 14 | 16 | 16 | 0.79 | 0.057 | 0.041 | 0.85 | 0.054 | 0.041 |
| 15 | 32 | 16 | 0.80 | 0.057 | 0.040 | 0.69 | 0.059 | 0.043 |
| 16 | 40 | 16 | 0.82 | 0.054 | 0.039 | 0.81 | 0.053 | 0.039 |
| 17 | 64 | 16 | 0.79 | 0.059 | 0.041 | 0.76 | 0.057 | 0.040 |
| 18 | 128 | 16 | 0.79 | 0.058 | 0.040 | 0.78 | 0.059 | 0.043 |
| 19 | 32 | 32 | 0.78 | 0.059 | 0.042 | 0.75 | 0.058 | 0.039 |
| 20 | 40 | 32 | 0.79 | 0.058 | 0.041 | 0.79 | 0.055 | 0.040 |
| 21 | 64 | 32 | 0.78 | 0.059 | 0.042 | 0.83 | 0.052 | 0.040 |
| 22 | 128 | 32 | 0.79 | 0.058 | 0.041 | 0.79 | 0.056 | 0.041 |
| 23 | 40 | 40 | 0.77 | 0.060 | 0.043 | 0.77 | 0.060 | 0.044 |
| 24 | 64 | 40 | 0.79 | 0.058 | 0.042 | 0.75 | 0.060 | 0.043 |
| 25 | 128 | 40 | 0.80 | 0.057 | 0.040 | 0.78 | 0.057 | 0.042 |
| 26 | 64 | 64 | 0.78 | 0.060 | 0.042 | 0.78 | 0.057 | 0.040 |
| 27 | 128 | 64 | 0.72 | 0.068 | 0.050 | 0.65 | 0.067 | 0.048 |
| 28 | 128 | 128 | 0.72 | 0.067 | 0.049 | 0.65 | 0.072 | 0.051 |

II. 160-163 of the revised manuscript

*(Dai and Trenberth, 2002). As a reference, the performance of some empirical approaches, including MLR, multi-variate nonlinear regression (MNR), decision tree, random forest, and Support Vector Machine (SVM) regression, was shown in Table 3. Clearly, the ANN model showed better performance than other tested approaches using same input variables (Table 2, Model#10).*

**Table 3 (revised): Model comparison between traditional empirical methods (MLR and MNR) and mechine-learning based empirical methods (Decision tree, Random Forest, and SVM). The statistics was derived from confimatory dataset (training data independent validation data) using input variables: T, S, DO, N, P, and Si. Note $R^2$ statistics in our study was based on the calculation of coefficient of determination, therefore negative $R^2$ could be derived if there were strong bias.**

| Model | Kernel Function | Input variables | RMSE | $R^2$ | MAE |
|---|---|---|---|---|---|
| MLR | - | T, S, DO, N, P, Si | 0.078 | 0.56 | 0.062 |
| MNR | - | T, S, DO, N, P, Si | 0.060 | 0.74 | 0.047 |
| Decision Tree | Simple Tree | T, S, DO, N, P, Si | 0.064 | 0.71 | 0.047 |
| | Medium Tree | T, S, DO, N, P, Si | 0.060 | 0.74 | 0.044 |
| | Complex Tree | T, S, DO, N, P, Si | 0.061 | 0.73 | 0.043 |
| Random Forest | Boosted Trees | T, S, DO, N, P, Si | 0.340 | -7.51 | 0.339 |
| | Bagged Trees | T, S, DO, N, P, Si | 0.056 | 0.77 | 0.04 |
| SVM | Linear | T, S, DO, N, P, Si | 0.079 | 0.55 | 0.061 |
| | Quadratic | T, S, DO, N, P, Si | 0.061 | 0.73 | 0.046 |
| | Cubic | T, S, DO, N, P, Si | 0.060 | 0.74 | 0.043 |
| | Fine Gaussian | T, S, DO, N, P, Si | 0.064 | 0.70 | 0.042 |
| | Medium Gaussian | T, S, DO, N, P, Si | 0.054 | 0.79 | 0.041 |
| | Coarse Gaussian | T, S, DO, N, P, Si | 0.069 | 0.65 | 0.054 |

**Referee comment by Richard Mills**

2. **Second, the authors do a good job of citing other papers in which authors have used similar ANN approaches for similar biogeochemical prediction tasks in marine waters, and compare the RMSE of their model with published values from other models. I think that the paper would be greatly improved if the authors could do a direct comparison. For instance, the authors cite the CANYON neural network model of Sauzede et al., 2017, which has been developed for the global ocean, but note that "coastal seas tend to show greater temporal and spatial variability than open oceans", which I believe is an argument for why they developed the model presented in their paper. I can easily imagine that the model presented here will outperform the CANYON model for prediction on the East China Sea shelf, but I think it would be interesting for the authors to demonstrate this: The CANYON model appears to be freely available online, and it would be interesting to see how much better a model trained speficially for the East China Sea shelf will outperform one developed for the global ocean.**

We would like to thank the referee very much for his suggestion. We applied the CANYON model developed by Sauzède et al. (2017) to exploratory dataset, result showed that the ANN model presented here outperformed the CANYON model developed for the global ocean for predicting $pH_T$ on the ECS shelf. We will add the corresponding information in the revised manuscript—as follows II. 185-192 of the revised manuscript:

*Sauzède et al. (2017) developed a neural network method to estimate pH with RMSE of 0.02 in the global ocean. Here the CANYON model developed by Sauzède et al. (2017) was applied to exploratory dataset (Fig. 8b), showed a RMSE of 0.09 and MAE of 0.06. It demonstrated that the ANN model presented here outperformed the CANYON model developed for the global ocean for predicting $pH_T$ on the ECS shelf, where carbon chemistry parameters are not only under the direct impact of Taiwan Warm Current and remote control of the Kuroshio water intrusion into the shelf but also significantly controlled by seasonal variations of the Changjiang discharge (e.g., Isobe and Matsuno, 2008; Chen et al., 2008; Chou et al., 2009). Taking into account the highly complex hydrographic, biological and chemical conditions, the accuracy of $pH_T$ presented is promising.*

[Figure]

**Figure 8 (revised): Comparison of retrieved pH$_T$ with corresponding observations for exploratory dataset. (a)-pH$_T$ retrieved by the ANN model vs observations; (b)-pH$_T$ retrieved by CANYON (Sauzède et al., 2017) vs observations. The red circles represent March 2018, the blue squares represent July 2018, the green triangles represent October 2018. The 1:1 line is shown in the plot as visual reference. Three statistics approaches used are the mean absolute error (MAE), the coefficient of determination (R$^2$), and the root mean squared error (RMSE). N represents the number of data points.**

**Referee comment by Richard Mills**

3. **Finally, the authors perform an intersting study in which they use prognostic variables from the Changjian Biology Finite-Volume Coastal Ocean Model (FVCOM) as input to their ANN model in order to recover the pH$_T$. I am not a marine biogeochemistry modeler, so perhaps I am missing something obvious, but I am guessing that mechanistic models like FVCOM can provide prognostic pH$_T$. Is this available from the FVCOM runs that were used, or could it be obtained using FVCOM, or ROMS, or another, similar model? If so, how would the prognostic pH$_T$ from FVCOM (or similar) compare to the pH$_T$ from the authors' own ANN model? And what is the motivation for using the ANN? Is it because it can potentially provide a more accurate pH$_T$, or because it can provide pH$_T$ for situations in which it is not desirable to run a forward simulation or reanalysis to get the pH$_T$, or some other reason? This may be obvious to an marine biogeochemist, but I and many of the readers of GMD don't have this expertise. The motivation needs to be explained for the general GMD audience.**

We fully agree with the referee. We will compared the prognostic pH$_T$ from FVCOM with retrieved pH$_T$ from our ANN model and add the following sentence to the paragraph in II. 214-218 of the revised manuscript:

*We also compared monthly average pH$_T$ provided by the Changjiang Biology FVCOM with pH$_T$ retrieved by the ANN model using the Changjiang Biology FVCOM output at the surface and bottom on the ECS shelf (Fig. 3S), it showed that the ANN model can potentially provide a more accurate pH$_T$. The possible reason was that the carbonate system from the Changjiang Biology FVCOM was not optimized due to challenges obtaining sufficient boundary information.*

[Figure]

**Figure S3 (revised): Comparison of monthly average pH$_T$ on the East China Sea shelf. Blue solid line represents retrieved pH$_T$ by the ANN model using Changjiang Biology FVCOM output; green solid line represents simulated pH$_T$ by the Changjiang Biology FVCOM. (a)-surface; (b)-bottom.**

**Referee comment by Richard Mills**

**4.  Detailed comments:**

**Lines 34-35: The authors state, while comparing ANNs to multiple linear regression, that ANNs have the advantage of not requiring 'an a priori model but rather "learn" the model from existing data'. I think it would be more precise to say that they are nonparametric models and do not require assuming any underlying statistical distribution.**

We agree with the referee. See II. 37-38 of the revised manuscript:

 *they are non-parametric models and do not require assuming any underlying statistical distribution.*

**Lines 75 and 78: The authors say that samples were "poisoned" by addition of HgCl2. I think it may be more idiomatic to say "sterilized".**

We agree with the referee. See Lines 80 and 87 of the revised manuscript:

 *sterilized*

**Line 81: "The final number of data used by the ANN model was 1854". I would say the final number of "observations" or "records", to be precise.**

We agree with the referee. See Line 94 of the revised manuscript:

*the final number of observations from confirmatory dataset was 1854*

**Line 94: The authors talk about a model being "over-matched". I believe that "overfitted" is the term they mean.**

We agree with the referee. See Line 107 of the revised manuscript:

*the testing set was used to monitor whether the model was over-**fitted*

**There are problems with low resolution for all of the figures. Figure 1 is not really even readable. Figures need to be re-generated with much higher resolution, or using vector, rather than raster, formats.**

We agree with the referee. Figures will be re-generated with higher resolution or using vector. As follows:

[Figure]

**Figure 1 (revised): Sampling stations during 11 cruises (as confirmatory dataset) from 2013 to 2017 on the East China Sea shelf.**

[Figure]

**Figure 2 (revised): Sampling stations for three cruises (as exploratory dataset) used to extend the utility of the ANN model. The green circles represent March 2018, the purple squares represent July 2018, the red triangles represent October 2018.**

[Figure]

**Figure 3 (revised): Schematic representation of the neural network algorithm to retrieve pH$_T$. (a)-the architecture of the ANN model. Input variables are observed temperature, salinity, dissolved oxygen, nitrate, phosphate, and silicate together with the geolocation (longitude and latitude) and time (month) of sampling; (b)-data distribution diagram for training and prediction.**

[Figure]

**Figure 4 (revised): Comparison of the performance of one hidden layer and two hidden layers on independent validation data. The result displayed are the mean and standard deviation of ten-fold cross-validation for each number of neurons in the hidden layer. The number in parentheses presents the number of neurons in the second hidden layer for two hidden layers.**

[Figure]

**Figure 5 (revised): Comparison of the performance of different training functions and transfer functions on independent validation data. (a)-three training functions: Gradient descent backpropagation (trainGD), Levenberg-Marquardt backpropagation (trainLM), Scaled conjugate gradient backpropagation (trainSCG); (b) three transfer functions: Log-sigmoid transfer function (logsig), Hyperbolic tangent sigmoid transfer function (tansig); Positive linear transfer function (poslin). The result displayed are the mean and standard deviation of ten-fold cross-validation for each number of neurons in the hidden layer.**

[Figure]

**Figure 6 (revised): Comparison of pH_T retrieved by the ANN model with corresponding observations. (a)-Training data (90% of confirmatory dataset); (b)-Independent validation data (10% of confirmatory dataset); (c)-Histogram of residuals for confirmatory dataset. The 1:1 line is shown in each plot as visual reference. Three statistics are the mean absolute error (MAE), the coefficient of determination ($R^2$), and the root mean squared error (RMSE). N represents the number of data points.**

[Figure]

**Figure 7 (revised): Box plots of the differences between retrieved pH_T minus the observations. (a)-the differences vs longitude (Mean±SE); (b)-the differences vs latitude (Mean±SE). The height of each box represents the mean value of the differences, the whisker represents the standard error (SE) value of the differences.**

[Figure]

**Figure 8 (revised): Comparison of retrieved pH$_T$ with corresponding observations for exploratory dataset. (a)-pH$_T$ retrieved by the ANN model vs observations; (b)-pH$_T$ retrieved by CANYON (Sauzède et al., 2017) vs observations. The red circles represent March 2018, the blue squares represent July 2018, the green triangles represent October 2018. The 1:1 line is shown in the plot as visual reference. Three statistics approaches used are the mean absolute error (MAE), the coefficient of determination (R$^2$), and the root mean squared error (RMSE). N represents the number of data points.**

[Figure]

**Figure 9 (revised): Sensitivity of the ANN model for environmental input variables: temperature (T), salinity (S), dissolved oxygen (DO), nitrate (N), phosphate (P) and silicate (Si).**

[Figure]

**Figure 10 (revised): Comparison of surface pH$_T$ retrieved by the ANN model using Changjiang Biology FVCOM output with corresponding observations at six sites repeated sampling for 3 to 4 years. Red dots represent observations pH$_T$, blue solid line represents retrieved pH$_T$, black dotted line represent retrieved pH$_T$ ± RMSE. (a)-station A1-5; (b)-station A1-6; (c)-station A6-7; (d)-station A6-9; (e)-station A7-5; (f)-station A8-5.**

[Figure]

**Figure 11 (revised): Comparison of water column pH$_T$ retrieved by the ANN model using Changjiang Biology FVCOM output with corresponding observations at six sites repeated sampling for 3 to 4 years. The 1:1 line is shown in the plot as a visual reference. Skill statistics include the mean absolute error (MAE), the coefficient of determination (R$^2$), and the root mean squared error (RMSE). N represents the number of data points.**

[Figure]

**Figure 12 (revised): Comparison of monthly average $pH_T$ on the East China Sea shelf. Blue solid line represents retrieved $pH_T$ by the ANN model using Changjiang Biology FVCOM output; black dotted line represents retrieved $pH_T \pm$ RMSE; red points show monthly-average $pH_T$ observations from 2013 to 2016. (a)-surface; (b)-bottom.**

[Figure]

**Figure S1 (revised): Comparison of three transfer functions. (a)-Log-sigmoid transfer function (logsig); (b) Hyperbolic tangent sigmoid transfer function (tansig); (c)-Positive linear transfer function (poslin).**

[Figure]

**Figure S2 (revised): Comparison of monthly-average environmental variables from the Changjiang Biology FVCOM with the corresponding observations at the surface and bottom on the East China Sea shelf. Blue and green solid lines represent surface and bottom simulated data from the Changjiang Biology FVCOM, respectively; red and black points show surface and bottom observation data from 2013 to 2016, respectively. (a)-temperature; (b)-salinity; (c)-dissolved oxygen; (d)-nitrate; (e)-phosphate; (f)-silicate.**

[Figure]

**Figure S3 (revised): Comparison of monthly average $pH_T$ on the East China Sea shelf. Blue solid line represents retrieved $pH_T$ by the ANN model using Changjiang Biology FVCOM output; green solid line represents simulated $pH_T$ by the Changjiang Biology FVCOM. (a)-surface; (b)-bottom.**

[revised manuscript text omitted]

210  We compared monthly input variables from the Changjiang Biology FVCOM with the corresponding observations at the surface and bottom (Fig. S2), it was found that simulated T and S were close to the observed value, simulated DO was high than the measured value at the bottom, simulated nutrients were high than the observed value at the surface. We also compared monthly average $pH_T$ provided by the Changjiang Biology

215 FVCOM with $pH_T$ retrieved by the ANN model using the Changjiang Biology FVCOM output at the surface and bottom on the ECS shelf (Fig. 3S), it showed that the ANN model can potentially provide a more accurate $pH_T$. The possible reason was that the carbonate system from the Changjiang Biology FVCOM was not optimized due to challenges obtaining sufficient boundary information.

Considering the discreteness and discontinuity of the sampling sites, we compared  $pH_T$ retrieved by the ANN model

220 using the Changjiang Biology FVCOM output with the corresponding observations at some sites with repeated sampling for 3 to 4 years. These sites were A1-5 (123.0140°E, 32.2145°N), A1-6 (123.2750°E, 32.2679°N), A6-7 (122.9880°E, 30.7050°N), A6-9 (123.4990°E, 30.5723°N), A7-5 (123.4990°E, 30.2523°N), and A8-5 (123.4930°E, 29.9940°N). Overall, the retrieved $pH_T$  agrees well with the observed values at the surface, except for three samples in summer (Fig. 10). There are relatively large deviations (greater than the RMSE of 0.04) in August 2013 at station A1-5

225 and A6-9, and in July 2016 at station A8-5.  To illustrate the application performance in the water column, a scatterplot of retrieved $pH_T$ vs observations at six sites with repeated sampling for 3 to 4 years (Fig. 11) shows that the ANN model predicts $pH_T$ with a RMSE of 0.05 and $R^2$ of 0.71.

We further compared monthly retrieved $pH_T$ by the ANN model using the Changjiang Biology FVCOM output with

230  in situ measured $pH_T$ values (Figure 12). The agreement is good here in winter, but large deviations appear in summer. The reduced performance in summer can be attributed in large part a reduced performance of the FVCOM model in predicting summertime input variables DO and S (see Figure S3).

**4 Summary and conclusions **

235 We have developed an artificial neural network model, demonstrated its reliability, and used it to retrieve monthly $pH_T$ for the period 2000-2016 on the East China Sea shelf.  We trained this ANN model using 11 cruise datasets from 2013 to 2017. In order to choose the optimal architecture of the ANN model, we tested different training and transfer functions, the number of neurons in two hidden layers, and different combinations of input

240 variables. We also validated the reliability of the ANN model with a root mean square error accuracy of 0.04 using three cruises in 2018 as exploratory dataset. The environmental input variable with the greatest sensitivity was dissolved oxygen, followed by salinity and temperature. We retrieved monthly $pH_T$ by the ANN model using 
[revised manuscript text omitted]
 and two hidden layers on independent validation data. The result displayed are the mean and standard deviation of ten-fold cross-validation for each number of neurons in the hidden layer. The number in parentheses presents the number of neurons in the second hidden layer for two hidden layers.**

440

[Figure]

[Figure]

**Figure 5: Comparison of the performance of different training functions and transfer functions on independent validation data. (a)- three training functions: Gradient descent backpropagation (trainGD), Levenberg-Marquardt backpropagation (trainLM), Scaled conjugate gradient backpropagation (trainSCG); (b) three transfer functions: Log-sigmoid transfer function (logsig), Hyperbolic tangent sigmoid transfer function (tansig); Positive linear transfer function (poslin). The result displayed are the mean and standard deviation of ten-fold cross-validation for each number of neurons in the hidden layer.**

[revised manuscript text omitted]

**Supplementary material**

**Table S1: The performance of different number of neurons for two hidden layers in the training step. Three statistics are the coefficient of determination ($R^2$), the root mean squared error (RMSE), and the mean absolute error (MAE).**

| Model | Number of neurons | | Training data | | | Independent validation data | | |
|---|---|---|---|---|---|---|---|---|
| | first hidden | second hidden | $R^2$ | RMSE | MAE | $R^2$ | RMSE | MAE |
| 1 | 4 | 4 | 0.68 | 0.071 | 0.054 | 0.67 | 0.072 | 0.057 |
| 2 | 8 | 4 | 0.70 | 0.070 | 0.050 | 0.67 | 0.069 | 0.050 |
| 3 | 16 | 4 | 0.76 | 0.062 | 0.045 | 0.76 | 0.062 | 0.045 |
| 4 | 32 | 4 | 0.74 | 0.063 | 0.046 | 0.79 | 0.062 | 0.048 |
| 5 | 40 | 4 | 0.76 | 0.062 | 0.044 | 0.76 | 0.061 | 0.045 |
| 6 | 64 | 4 | 0.79 | 0.058 | 0.041 | 0.78 | 0.056 | 0.043 |
| 7 | 128 | 4 | 0.76 | 0.062 | 0.045 | 0.74 | 0.062 | 0.044 |
| 8 | 8 | 8 | 0.73 | 0.065 | 0.047 | 0.73 | 0.065 | 0.048 |

| | | | | | | | | |
|---|---|---|---|---|---|---|---|---|
| 9 | 16 | 8 | 0.78 | 0.059 | 0.042 | 0.78 | 0.058 | 0.044 |
| 10 | 32 | 8 | 0.78 | 0.059 | 0.042 | 0.83 | 0.053 | 0.039 |
| 11 | 40 | 8 | 0.79 | 0.059 | 0.042 | 0.77 | 0.055 | 0.040 |
| 12 | 64 | 8 | 0.77 | 0.061 | 0.044 | 0.76 | 0.059 | 0.042 |
| 13 | 128 | 8 | 0.77 | 0.060 | 0.042 | 0.79 | 0.059 | 0.043 |
| 14 | 16 | 16 | 0.79 | 0.057 | 0.041 | 0.85 | 0.054 | 0.041 |
| 15 | 32 | 16 | 0.80 | 0.057 | 0.040 | 0.69 | 0.059 | 0.043 |
| 16 | 40 | 16 | 0.82 | 0.054 | 0.039 | 0.81 | 0.053 | 0.039 |
| 17 | 64 | 16 | 0.79 | 0.059 | 0.041 | 0.76 | 0.057 | 0.040 |
| 18 | 128 | 16 | 0.79 | 0.058 | 0.040 | 0.78 | 0.059 | 0.043 |
| 19 | 32 | 32 | 0.78 | 0.059 | 0.042 | 0.75 | 0.058 | 0.039 |
| 20 | 40 | 32 | 0.79 | 0.058 | 0.041 | 0.79 | 0.055 | 0.040 |
| 21 | 64 | 32 | 0.78 | 0.059 | 0.042 | 0.83 | 0.052 | 0.040 |
| 22 | 128 | 32 | 0.79 | 0.058 | 0.041 | 0.79 | 0.056 | 0.041 |
| 23 | 40 | 40 | 0.77 | 0.060 | 0.043 | 0.77 | 0.060 | 0.044 |
| 24 | 64 | 40 | 0.79 | 0.058 | 0.042 | 0.75 | 0.060 | 0.043 |
| 25 | 128 | 40 | 0.80 | 0.057 | 0.040 | 0.78 | 0.057 | 0.042 |
| 26 | 64 | 64 | 0.78 | 0.060 | 0.042 | 0.78 | 0.057 | 0.040 |
| 27 | 128 | 64 | 0.72 | 0.068 | 0.050 | 0.65 | 0.067 | 0.048 |
| 28 | 128 | 128 | 0.72 | 0.067 | 0.049 | 0.65 | 0.072 | 0.051 |

[Figure]

**Figure S1: Comparison of three transfer functions. (a)-Log-sigmoid transfer function (logsig); (b) Hyperbolic tangent sigmoid transfer function (tansig); (c)-Positive linear transfer function (poslin).**

495

[Figure]

**Figure S2: Comparison of monthly-average environmental variables from the Changjiang Biology FVCOM with the corresponding observations at the surface and bottom on the East China Sea shelf. Blue and green solid lines represent surface and bottom simulated data from the Changjiang Biology FVCOM, respectively; red and black points show surface and bottom observation data from 2013 to 2016, respectively. (a)-temperature; (b)-salinity; (c)-dissolved oxygen; (d)-nitrate; (e)-phosphate; (f)-silicate.**

500

[Figure]

**Figure S3: Comparison of monthly average pH$_T$ on the East China Sea shelf. Blue solid line represents retrieved pH$_T$ by the ANN model using Changjiang Biology FVCOM output; green solid line represents simulated pH$_T$ by the Changjiang Biology FVCOM. (a)-surface; (b)-bottom.**

---

## Referee Comment (RC2) · Anonymous Referee #2 · 14 Jul 2020

Manuscript by Li et. al. presents a Artificial Neural Network (ANN) approach for modeling pH in East China Sea using observations collected from cruises. They have also applied their trained model to prognostic outputs from FVCOM model simulation.

Study is well designed and described and manuscript is easy to read and follow. Below are few questions/comments on the article that would help improve the clarity of the paper. A number of my questions were already captured by Reviewer 1 and addressed well by Author's response, so I will skip some of those in my review.

Choice of inputs: A total of nine variables were used as inputs to the ANN, six of which were direct measurements (T, S, DO, N, P, and Si). Lines 105-107 notes "We found

geographical information to be a powerful addition in improving the skill of the method (see Table 2), allowing the network to learn spatio-temporal patterns that could not be explained by other input variables (Sasse et al., 2013)."

Adding geographical information does appear to improve the performance for the initial model training (Table 2). However, the cruise tracks are only sampling certain latitudes lead to a biased sampling. BUT can this lead to a geographically biased training? This bias may not be apparent even in the validation using data from three cruises, since they too are in same bands as before. But when applied to data from FVCOM, there are biases reported in Figure 5. Is it possible that the model is not generalized enough for other regions?

Lines 100-109 explains the choice of variables for all but one variable "month". I assume the variable was added to capture the seasonality. However, a significant bias was still reported in August 2013, and July 2016. These biases are being attributed to sudden increase in the river discharge, but did that not affect July 2014, 2015, 2017? What is the role "month" is playing in the ANN model? Once trained is the expectation for the model to be able to interpolate between the month when the samples were not taken?

ANN application to FVCOM: Inputs to the ANN models training, based on cruise observations, were instantaneous measurements. What was the spatial resolution, time step and temporal output frequency from FVCOM model to provide comparable outputs. If monthly averages were used, please comment on applicability and validity of applying model trained based on instantaneous measurements to monthly averages?

Application to FVCOM, scales the model to extended space and time, which I think is a key strength and contribution of this work. Spatial bias has been discussed and reported in the manuscript, but it would be important to discuss the model performance in time. Cruise observations were only from select few months, but is the model able to fill in between the seasons reasonably? And if yes, why? If no, why not?
Final output of ANN applied to FVCOM data would be a time series of full spatial data i.e. pixel-wise pH estimate for ECS. That product is a key contribution that should be included in the manuscript, and spatial and temporal patterns of the outputs should be discussed.

Variable importance in the ANN model: The methodology here is not clear to me. What does adding 5% to environmental variable separately means? Is this a perturbation to the data to test its sensitivity? In either case, I am not at all convinced that this can be quantified as variable importance. There also is mention of "variable with greatest weight was DO, followed by S and T". What weights are looking at here, is this from the final trained model? From first layer, from second layer, or both? This section need additional detail and discussion to convey and convince the interpretation of variable importance.

Please improve the quality of figures 1 and 6, as they are difficult to read and follow.

In Figures 9 and 11, I am unable to understand what "ANN Model - RMSE" and "ANN Model + RMSE", and thus the related discussions.

---

## Author Response (AR1)

**Final response in the interactive discussion**

Dear Referees, dear Editor,

We would like to thank you very much for your positive comments and constructive suggestions to our manuscript "*Retrieving monthly and interannual pH_T on the East China Sea shelf using an artificial neural network: ANN-pH_T-v1*".

In this document, we would like to provide our responses to the comments of each of the referees in one single document and to outline the corresponding changes to the manuscript. We will represent the referee comment in **bold** font, and our response in normal font. Quotations from the original manuscript will be in *italics*, changes as part of the manuscript revision will be highlighted as underlined. For the sake of clarity and brevity, we have omitted the introductory parts of the referee report (this omittance is marked as [...]).

We hope that our response together with the revision of the manuscript sufficiently addresses the referee' concerns.

Sincerely,

Xiaoshuang Li (on behalf of the author team)

**Referee comment #1 (by Richard Mills)**

**[…] I came away from my reading of the paper with the following major questions/concerns which, if addressed, will greatly improve the quality of the paper:**

1. **First, since the paper has been submitted to a model development journal, I would like to see more information on how and why the authors arrived at the particular form of the machine-learning model they used, and how this model performed against some other possible model architectures. The authors have used a feed-forward multilayer perceptron network with two hidden layers (with 40 neurons in the first layer and 16 in the second) and full connectivity between the layers. Why did the authors decide on two layers, and how did they choose the number of neurons in each layer? (They do state that they tried varying the number of neurons in each layer, but don't give further details.) How did they choose the activation function? And why did they choose a neural network, instead of another approach such as k-nearest neighbors, random forest regression, or support vector regression? When I first started working in machine learning, around two decades ago, it would not have been expected for authors to try a variety of different types of models, as this would likely involve substantial code development effort, as well as possibly significant computational expense for training models. Today, however, it is easy to try many different models, as code provided in many easily obtained packages such as Scikit-learn or those provided by Matlab (the environment that the authors use for this study), and it is becoming the norm for papers presenting the development of machine-learning models to compare several types to determine the one that performs best for the chosen task. I would like to see some comparison against other models (some of the ones easily constructed using Matlab) to demonstrate that the ANN is the most appropriate choice.**

We thank the referee for the suggestion: the required details should, in fact, be provided to the reader. We will add, in the revised manuscript, the corresponding information (Why did the authors decide on two layers, and how did they choose the number of neurons in each layer? How did they choose the activation function? And why did they choose a neural network, instead of another approach such as k-nearest neighbors, random forest regression, or support vector regression?)—as follows ll. 111-127 and 160-163 of the revised manuscript:

ll. 111-127 of the revised manuscript:

*In our study, calculations were done in the MathWorks Matlab environment, using the Deep Learning Toolbox.*
*First, we compared the performance of one hidden layer vs. two hidden layers in predicting independent validation data. The number of neurons varied from $2^2$ to $2^8$ for the first hidden layer and was fixed at four in the second hidden layer for the two*

*hidden layers model; the number of neurons in the first layer was the same in the one hidden layer vs. two hidden layers model (Fig. 4). The ten-fold cross-validation showed that the model with two hidden layers performed better as the number of neurons increased. Second, in order to choose suitable training techniques and activation functions of the ANN model with two hidden layers, we tested three training functions (Gradient descent backpropagation (trainGD), Levenberg-Marquardt backpropagation (trainLM), and Scaled conjugate gradient backpropagation (trainSCG)), which differed in how the weights are modified, and three transfer functions (Log-sigmoid transfer function (logsig), Hyperbolic tangent sigmoid transfer function (tansig), and Positive linear transfer function (poslin)) (Fig. 5). The output values of logsig, tansig and poslin were compressed onto [0, 1], [-1, 1], and [0, +∞], respectively (Fig. S1). As the number of neurons increased, the performances of trainGD and tansig became poor. Although there was no obvious difference between trainLM and trainSCG, the training technique trainSCG was selected and the transfer function logsig was applied to two hidden layers considering the overall performance (Fig. 5). Third, in the training phase of the ANN model, the number of neurons was tested, varying from 4 to 128 for two hidden layers (Table S1). Best performance for both training data and independent validation data was obtained with 40 neurons in the first hidden layer and 16 neurons in the second layer. Finally, different combinations of input variables were tested to choose the optimal architecture of the ANN model (Table 2); best performance was obtained using longitude, latitude, month, T, S, DO, N, P and Si as input variables.*

[Figure]

*Figure 4 (revised): Comparison of the performance of one hidden layer vs. two hidden layers in predicting independent validation data. The number of neurons in the first hidden layer was the same in the one hidden layer vs. two hidden layers model, numbers in parentheses show the number of neurons in the second hidden layer (for the two hidden layers model). Bars show the mean and standard deviation of the Root-Mean-Square-Error over a ten-fold cross-validation, for different numbers of neurons in the first hidden layer.*

[Figure]

[Figure]

*Figure 5 (revised): Comparison of the performance of different training functions and transfer functions on independent validation data. (a)-three training functions: Gradient descent backpropagation (trainGD), Levenberg-Marquardt backpropagation (trainLM), and Scaled conjugate gradient backpropagation (trainSCG); (b) three transfer functions: Log-sigmoid transfer function (logsig), Hyperbolic tangent sigmoid transfer function (tansig), and Positive linear transfer function (poslin). Bars show the mean and standard deviation of the Root-Mean-Square-Error over a ten-fold cross-validation, for different numbers of neurons in the first hidden layer.*

[Figure]

*Figure S1 (revised): Comparison of three transfer functions. (a)-Log-sigmoid transfer function (logsig); (b) Hyperbolic tangent sigmoid transfer function (tansig); (c)-Positive linear transfer function (poslin).*

*Table S1 (revised): The performance of different number of neurons for two hidden layers in the training step. Three statistics are the coefficient of determination ($R^2$), the root mean squared error (RMSE), and the mean absolute error (MAE).*

| Model | Number of neurons | | Training data | | | Independent validation data | | |
|---|---|---|---|---|---|---|---|---|
| | first hidden | second hidden | $R^2$ | RMSE | MAE | $R^2$ | RMSE | MAE |
| 1 | 4 | 4 | 0.68 | 0.071 | 0.054 | 0.67 | 0.072 | 0.057 |
| 2 | 8 | 4 | 0.70 | 0.070 | 0.050 | 0.67 | 0.069 | 0.050 |
| 3 | 16 | 4 | 0.76 | 0.062 | 0.045 | 0.76 | 0.062 | 0.045 |
| 4 | 32 | 4 | 0.74 | 0.063 | 0.046 | 0.79 | 0.062 | 0.048 |
| 5 | 40 | 4 | 0.76 | 0.062 | 0.044 | 0.76 | 0.061 | 0.045 |
| 6 | 64 | 4 | 0.79 | 0.058 | 0.041 | 0.78 | 0.056 | 0.043 |
| 7 | 128 | 4 | 0.76 | 0.062 | 0.045 | 0.74 | 0.062 | 0.044 |
| 8 | 8 | 8 | 0.73 | 0.065 | 0.047 | 0.73 | 0.065 | 0.048 |
| 9 | 16 | 8 | 0.78 | 0.059 | 0.042 | 0.78 | 0.058 | 0.044 |

| 10 | 32 | 8 | 0.78 | 0.059 | 0.042 | 0.83 | 0.053 | 0.039 |
| 11 | 40 | 8 | 0.79 | 0.059 | 0.042 | 0.77 | 0.055 | 0.040 |
| 12 | 64 | 8 | 0.77 | 0.061 | 0.044 | 0.76 | 0.059 | 0.042 |
| 13 | 128 | 8 | 0.77 | 0.060 | 0.042 | 0.79 | 0.059 | 0.043 |
| 14 | 16 | 16 | 0.79 | 0.057 | 0.041 | 0.85 | 0.054 | 0.041 |
| 15 | 32 | 16 | 0.80 | 0.057 | 0.040 | 0.69 | 0.059 | 0.043 |
| 16 | 40 | 16 | 0.82 | 0.054 | 0.039 | 0.81 | 0.053 | 0.039 |
| 17 | 64 | 16 | 0.79 | 0.059 | 0.041 | 0.76 | 0.057 | 0.040 |
| 18 | 128 | 16 | 0.79 | 0.058 | 0.040 | 0.78 | 0.059 | 0.043 |
| 19 | 32 | 32 | 0.78 | 0.059 | 0.042 | 0.75 | 0.058 | 0.039 |
| 20 | 40 | 32 | 0.79 | 0.058 | 0.041 | 0.79 | 0.055 | 0.040 |
| 21 | 64 | 32 | 0.78 | 0.059 | 0.042 | 0.83 | 0.052 | 0.040 |
| 22 | 128 | 32 | 0.79 | 0.058 | 0.041 | 0.79 | 0.056 | 0.041 |
| 23 | 40 | 40 | 0.77 | 0.060 | 0.043 | 0.77 | 0.060 | 0.044 |
| 24 | 64 | 40 | 0.79 | 0.058 | 0.042 | 0.75 | 0.060 | 0.043 |
| 25 | 128 | 40 | 0.80 | 0.057 | 0.040 | 0.78 | 0.057 | 0.042 |
| 26 | 64 | 64 | 0.78 | 0.060 | 0.042 | 0.78 | 0.057 | 0.040 |
| 27 | 128 | 64 | 0.72 | 0.068 | 0.050 | 0.65 | 0.067 | 0.048 |
| 28 | 128 | 128 | 0.72 | 0.067 | 0.049 | 0.65 | 0.072 | 0.051 |

II. 160-163 of the revised manuscript:

*(Dai and Trenberth, 2002). As a reference, the performance of some other empirical approaches, including MLR, multi-variate nonlinear regression (MNR), decision tree, random forest, and Support Vector Machine (SVM) regression, is shown in Table 3. The selected ANN model (Table 2, Model#10) showed better performance than the other tested approaches using the same input variables (Table 3).*

*Table 3 (revised): Model comparison between traditional empirical methods (MLR and MNR) and mechine-learning based empirical methods (Decision tree, Random Forest, and SVM). The statistics was derived from confimatory dataset (training data independent validation data) using input variables: T, S, DO, N, P, and Si. Note $R^2$ statistics in our study was based on the calculation of coefficient of determination, therefore negative $R^2$ could be derived if there were strong bias.*

| Model | Kernel Function | Input variables | RMSE | $R^2$ | MAE |
|---|---|---|---|---|---|
| MLR | - | T, S, DO, N, P, Si | 0.078 | 0.56 | 0.062 |
| MNR | - | T, S, DO, N, P, Si | 0.060 | 0.74 | 0.047 |
| Decision Tree | Simple Tree | T, S, DO, N, P, Si | 0.064 | 0.71 | 0.047 |
| | Medium Tree | T, S, DO, N, P, Si | 0.060 | 0.74 | 0.044 |
| | Complex Tree | T, S, DO, N, P, Si | 0.061 | 0.73 | 0.043 |
| Random Forest | Boosted Trees | T, S, DO, N, P, Si | 0.340 | -7.51 | 0.339 |
| | Bagged Trees | T, S, DO, N, P, Si | 0.056 | 0.77 | 0.04 |
| SVM | Linear | T, S, DO, N, P, Si | 0.079 | 0.55 | 0.061 |
| | Quadratic | T, S, DO, N, P, Si | 0.061 | 0.73 | 0.046 |
| | Cubic | T, S, DO, N, P, Si | 0.060 | 0.74 | 0.043 |
| | Fine Gaussian | T, S, DO, N, P, Si | 0.064 | 0.70 | 0.042 |
| | Medium Gaussian | T, S, DO, N, P, Si | 0.054 | 0.79 | 0.041 |
| | Coarse Gaussian | T, S, DO, N, P, Si | 0.069 | 0.65 | 0.054 |

**Referee comment #1 (by Richard Mills)**

2. **Second, the authors do a good job of citing other papers in which authors have used similar ANN approaches for similar biogeochemical prediction tasks in marine waters, and compare the RMSE of their model with published values from other models. I think that the paper would be greatly improved if the authors could do a direct comparison. For instance, the authors cite the CANYON neural network model of Sauzede et al., 2017, which has**

been developed for the global ocean, but note that "coastal seas tend to show greater temporal and spatial variability than open oceans", which I believe is an argument for why they developed the model presented in their paper. I can easily imagine that the model presented here will outperform the CANYON model for prediction on the East China Sea shelf, but I think it would be interesting for the authors to demonstrate this: The CANYON model appears to be freely available online, and it would be interesting to see how much better a model trained speficially for the East China Sea shelf will outperform one developed for the global ocean.

We would like to thank the referee very much for his suggestion. We applied the CANYON model developed by Sauzède et al. (2017) to exploratory dataset, result showed that the ANN model presented here outperformed the CANYON model developed for the global ocean for predicting $pH_T$ on the ECS shelf. We will add the corresponding information in the revised manuscript—as follows II. 187-194 of the revised manuscript:

*Sauzède et al. (2017) developed a neural network method to estimate pH with RMSE of 0.02 in the global ocean. As a further comparison we applied the CANYON model developed by Sauzède et al. (2017) to our coastal exploratory dataset (Fig. 8b), and obtained an RMSE of 0.09 and MAE of 0.06. It is not surprising that the ANN model (developed here for the ECS shelf) outperforms the CANYON model (developed for the global ocean) for predicting $pH_T$ on the ECS shelf. The carbon chemistry parameters in this region are not only under the direct impact of Taiwan Warm Current and remote control of the Kuroshio water intrusion into the shelf, but are also significantly controlled by seasonal variations of the Changjiang discharge (e.g., Isobe and Matsuno, 2008; Chen et al., 2008; Chou et al., 2009). Taking into account the highly complex hydrographic, biological and chemical conditions, the accuracy of $pH_T$ presented is promising.*

[Figure]

*Figure 8 (revised): Comparison of retrieved $pH_T$ with corresponding observations for exploratory dataset. (a)-$pH_T$ retrieved by the ANN model vs observations; (b)-$pH_T$ retrieved by CANYON (Sauzède et al., 2017) vs observations. The red circles represent March 2018, the blue squares represent July 2018, the green triangles represent October 2018. The 1:1 line is shown in the plot as visual reference. Three statistics approaches used are the mean absolute error (MAE), the coefficient of determination ($R^2$), and the root mean squared error (RMSE). N represents the number of data points.*

**Referee comment #1 (by Richard Mills)**

3. **Finally, the authors perform an intersting study in which they use prognostic variables from the Changjian Biology Finite-Volume Coastal Ocean Model (FVCOM) as input to their ANN model in order to recover the $pH_T$. I am not a marine biogeochemistry modeler, so perhaps I am missing something obvious, but I am guessing that mechanistic models like FVCOM can provide prognostic $pH_T$. Is this available from the FVCOM runs that were used, or could it be obtained using FVCOM, or ROMS, or another, similar model? If so, how would the prognostic $pH_T$ from FVCOM (or similar) compare to the $pH_T$ from the authors' own ANN model? And what is the motivation for using the ANN? Is it because it can potentially provide a more accurate $pH_T$, or because it can provide $pH_T$ for situations**

**in which it is not desirable to run a forward simulation or reanalysis to get the pH$_T$, or some other reason? This may be obvious to an marine biogeochemist, but I and many of the readers of GMD don't have this expertise. The motivation needs to be explained for the general GMD audience.**

We fully agree with the referee. We will compare the prognostic pH$_T$ from FVCOM with retrieved pH$_T$ from our ANN model and add the following sentence to the paragraph in ll. 222-225 of the revised manuscript:

*Comparisons of monthly average pH$_T$ from the Changjiang Biology FVCOM model with pH$_T$ retrieved by the ANN model suggested that the ANN model can potentially provide a more accurate pH$_T$ (Fig. S3). The possible reason was that the carbonate system from the Changjiang Biology FVCOM was not optimized due to challenges obtaining sufficient boundary information.*

[Figure]

*Figure S3 (revised): Comparison of monthly average pH$_T$ on the East China Sea shelf. Blue solid line represents retrieved pH$_T$ by the ANN model using Changjiang Biology FVCOM output; green solid line represents simulated pH$_T$ by the Changjiang Biology FVCOM; red points show monthly average pH$_T$ observations from 2013-2016. (a)-surface; (b)-bottom.*

**Referee comment #1 (by Richard Mills)**

**4.   Detailed comments:**
**Lines 34-35: The authors state, while comparing ANNs to multiple linear regression, that ANNs have the advantage of not requiring 'an a priori model but rather "learn" the model from existing data'. I think it would be more precise to say that they are nonparametric models and do not require assuming any underlying statistical distribution.**

We agree with the referee. See ll. 37-38 of the revised manuscript:
 may be a greater flexibility and versatility in modelling complex nonlinear relationships.

**Lines 75 and 78: The authors say that samples were "poisoned" by addition of HgCl2. I think it may be more idiomatic to say "sterilized".**

We agree with the referee. See Lines 79 and 86 of the revised manuscript:

 *sterilized*

**Line 81: "The final number of data used by the ANN model was 1854". I would say the final number of "observations" or "records", to be precise.**

We agree with the referee. See Lines 93-94 of the revised manuscript:

*The final number of observations in the confirmatory dataset was 1854 (see Table 1 for more detailed information on the field survey).*

**Line 94: The authors talk about a model being "over-matched". I believe that "overfitted" is the term they mean.**

We agree with the referee. See Line 107 of the revised manuscript:

*the testing set was used to monitor whether the model was over-fitted*

**Referee comment #2**

1. **Choice of inputs: A total of nine variables were used as inputs to the ANN, six of which were direct measurements (T, S, DO, N, P, and Si). Lines 105-107 notes "We found geographical information to be a powerful addition in improving the skill of the method (see Table 2), allowing the network to learn spatio-temporal patterns that could not be explained by other input variables (Sasse et al., 2013)." Adding geographical information does appear to improve the performance for the initial model training (Table 2). However, the cruise tracks are only sampling certain latitudes lead to a biased sampling. BUT can this lead to a geographically biased training? This bias may not be apparent even in the validation using data from three cruises, since they too are in same bands as before. But when applied to data from FVCOM, there are biases reported in Figure 5. Is it possible that the model is not generalized enough for other regions?**

We agree with the referee. The model is not generalized enough for other regions. This model was trained using cruises datasets on the ECS shelf, can be applied to retrieve $pH_T$ on the ECS shelf. But *this approach can be applied to other regions to predict pH by suitably adapting the input variables and network structure using local datasets*.

2. **Lines 100-109 explains the choice of variables for all but one variable "month". I assume the variable was added to capture the seasonality. However, a significant bias was still reported in August 2013, and July 2016. These biases are being attributed to sudden increase in the river discharge, but did that not affect July 2014, 2015, 2017? What is the role "month" is playing in the ANN model? Once trained is the expectation for the model to be able to interpolate between the month when the samples were not taken?**

In order to retrieve monthly pH$_T$, the monthly T, S, DO, N, P and Si from the Changjiang Biology Finite-Volume Coastal Ocean Model (FVCOM) (http://47.101.49.44/wms/demo) were fed into the ANN model as input variables. Here a significant bias was reported in August 2013 and July 2016. See Lines 229-232 of the revised manuscript:

*Overall, the retrieved pH$_T$ agrees well (within the ANN model accuracy: ANN±RMSE) with the observed values at the surface, except for three samples in summer (Fig. 10). There are relatively large deviations (greater than the RMSE of 0.04) in August 2013 at station A1-5 and A6-9, and in July 2016 at station A8-5.*

The reduced performance in summer can be attributed in large part a reduced performance of the Changjiang Biology FVCOM in predicting summertime input variables S, DO, and nutrients (Fig. S2). See Lines 219-222 of the revised manuscript:

*Comparisons of monthly-average FVCOM model variables with surface and bottom observations on the ECS shelf showed that simulated T was close to observed values (Fig. S2a), simulated S was also close to observed values except at the bottom in August 2013 and at the surface in July 2016 (Fig. S2b), simulated DO was higher than observed at the bottom (Fig. S2c), and simulated nutrients were higher than observed at the surface (Fig. S2d-S2f).*

[Figure]

*Figure S2: Comparison of monthly-average environmental variables from the Changjiang Biology FVCOM with the corresponding observations at the surface and bottom on the East China Sea shelf. Blue and green solid lines represent surface and bottom simulated data from the Changjiang Biology FVCOM, respectively; red and black points show surface and bottom observation data from 2013 to 2016, respectively. (a)-temperature; (b)-salinity; (c)-dissolved oxygen; (d)-nitrate; (e)-phosphate; (f)-silicate.*

The variable "month" was added to capture the seasonality. The reliability of the ANN model was evaluated using independent observations from 3 cruises (March, July, and October) in 2018, and showed a root mean square error accuracy of 0.04. The cruise dataset during October was not used in ANN model development.

3. **ANN application to FVCOM: Inputs to the ANN models training, based on cruise observations, were instantaneous measurements. What was the spatial resolution, time step and temporal output frequency from FVCOM model to provide comparable outputs. If monthly averages were used, please comment on applicability and validity of applying model trained based on instantaneous measurements to monthly averages?**

We agree with the referee. We will compare monthly input variables from FVCOM model with instantaneous observations and add the following sentence to the paragraph in II. 217-222 of the revised manuscript:

*The resolution of the Changjiang Biology FVCOM output is 1-10 km in the horizontal, 10 depth levels in the vertical, and day in the temporal (refered Ge et al., (2013) for detail information). Comparisons of monthly-average FVCOM model variables with surface and bottom observations on the ECS shelf showed that simulated T was close to observed values (Fig. S2a), simulated S was also close to observed values except at the bottom in August 2013 and at the surface in July 2016 (Fig. S2b), simulated DO was higher than observed at the bottom (Fig. S2c), and simulated nutrients were higher than observed at the surface (Fig. S2d-S2f).*

**4. Application to FVCOM, scales the model to extended space and time, which I think is a key strength and contribution of this work. Spatial bias has been discussed and reported in the manuscript, but it would be important to discuss the model performance in time. Cruise observations were only from select few months, but is the model able to fill in between the seasons reasonably? And if yes, why? If no, why not? Final output of ANN applied to FVCOM data would be a time series of full spatial data i.e. pixel-wise pH estimate for ECS. That product is a key contribution that should be included in the manuscript, and spatial and temporal patterns of the outputs should be discussed.**

We agree with the referee, we trained the ANN model only using few months observations. In order to assess the ability of the ANN model to fill in other months not used in ANN model development, we applied it to October (not used) in 2018 with a root mean square error accuracy of 0.04; we also added spatial and temporal patterns of ANN-derived $pH_T$. See Section 3.4.2 of the revised manuscript, as follows:

**3.4.2 Spatial and temporal patterns of ANN-derived $pH_T$**

*The temporal and spatial variations of monthly surface $pH_T$ from 2000-2016 based on Changjiang Biology FVCOM output are shown in Figure 13. During the dry season (November to March of the next year), $pH_T$ values vary from ~7.62 to ~8.24. Relatively higher $pH_T$ values are found in the southeastern of the study area (Chou et al., 2011), whereas lower $pH_T$ values are found in the northeastern of the study area. During the wet season (April to October), $pH_T$ values vary from ~7.77 to ~8.35, water of higher $pH_T$ corresponded well to the seasonal dispersion of the Changjiang Dilute Water (Chou et al., 2009, 2013). Water of higher $pH_T$ is found in the center of the study area during April, spreads to the southwestern part of the study area (along the coast of China) during May and June, shifts to the northeastern part of the study area during August. In September and October, water of higher $pH_T$ is found in the southeastern part of the study area, strongly influenced by the Taiwan Warm Current (Qu et al., 2015).*

*A clear seasonality is that surface $pH_T$ gradually increases during spring (March to May), after which it gradually decreases during summer and fall (June to November) (Fig. 14). The surface $pH_T$ displays its maximum in May and minimum in December, and the $pH_T$ varies seasonally by up to ~0.3 unit. Larger changes in pH were also discovered in the Washington Shelf, the pH varied ~1.0 unit over the seasons and ~1.5 unit spanning 8 years (Wootton et al., 2008). Accordingly, seasonal dynamics of surface $pH_T$ can be mainly attributed to temperature changes and strong biological activities (production and respiration processes) over the season. From March to June, a rapid increase in surface $pH_T$ indicates that production increases faster than respiration, which can be reflected in the drop in surface phosphate (Fig. S5d) and apparent oxygen utilization (AOU) (Fig. S5c). It may be driven by the Changjiang discharge (Fig. S4), which carries large amount of nutrients, result in stronger primary production in warm seasons under the combined action of nutrients and suitable temperature (Gong et al., 2011). From July to October, although surface temperature remains at a high level (Fig. S5a), the rise in surface AOU*

*(Fig. S5c) suggest a decrease in primary production or increase of respiration, which leads to a gradual drop in surface pH$_T$ (Wootton et al., 2012). It implies respiration processes dominate relative to primary production during summer and fall.*

[Figure]

*Figure 13: Spatial distribution of monthly average surface pH$_T$ retrieved by the ANN model using Changjiang Biology FVCOM output. (a)-January; (b)-February; (c)-March; (d)-April; (e)-May; (f)-June; (g)-July; (h)-August; (i)-September; (j)-October; (k)-November; (l)-December.*

[Figure]

*Figure 14: Seasonal cycles of surface pH$_T$ on the East China Sea shelf from 2000-2016. The green circles represent monthly regional average, the blue dashed represents mean value of each month.*

[Figure]

*Figure S4: Monthly average water discharge and its standard deviation (DaTong Station, data derived from the Hydrological Information Center of China, http://www.hydroinfo.gov.cn/).*

[Figure]

*Figure S5: Seasonal cycles of surface T (a), S (b), AOU (c), and P (d) from Changjiang Biology FVCOM output on the East China Sea shelf from 2000-2016. The green circles represent monthly regional average, the blue dashed represents mean value of each month.*

5. **Variable importance in the ANN model: The methodology here is not clear to me. What does adding 5% to environmental variable separately means? Is this a perturbation to the data to test its sensitivity? In either case, I am not at all convinced that this can be quantified as variable importance. There also is mention of "variable with greatest weight was DO, followed by S and T". What weights are looking at here, is this from the final trained model? From first layer, from second layer, or both? This section need additional detail and discussion to convey and convince the interpretation of variable importance.**

We agree with the referee. We only want to test the ANN model sensitivity to environmental input variables, and we re-wrote section 3.3. See Section 3.3 of the revised manuscript, as follows:

*3.3 ANN model sensitivity to environmental input variables*

*To assess the ANN model sensitivity to different environmental input variables, we added 5% perturbation for each environmental variable separately. Statistically, with 5% T errors added, the ANN model showed slight overestimation in $pH_T$, with mean bias (MB) of 0.0059, RMSE of 0.0079, and $R^2$ of 0.9949 (Fig. 9a); with 5% DO errors added, the ANN model also showed slight $pH_T$ overestimation, with MB of 0.0050, RMSE of 0.0090, and $R^2$ of 0.9934 (Fig. 9c); with 5% S errors added, the ANN model showed overestimation in pHT, with MB of -0.0111, RMSE of 0.0162, and $R^2$ of 0.9789 (Fig. 9b). These results suggested that the ANN model responded to T and DO errors in a positive way, S errors in a negative way. The positive response to increasing DO reflects positive correlation between $pH_T$ and DO (Cai et al., 2011), which can be attributed to the processes of photosynthesis (generating DO and removing $CO_2$, hence increasing pH) and aerobic respiration (consuming*

*DO and generating $CO_2$, hence lowering pH); the negative response to increasing S reflects the influence of the (lower salinity) Changjiang discharge, carrying large amounts of nutrients that fuel increased primary production (uptake of nutrients and $CO_2$, hence raising the pH) in surface waters during warm seasons (Gong et al., 2011). It was found that the ANN model was insensitive to nutrients errors (Fig. 9d-9f) and most sensitive to S errors (Fig. 9b), followed by DO and T errors.*

[Figure]

*Figure 9: Sensitivity of the ANN model for environmental input variables. (a)-temperature (T); (b) salinity (S); (c)-dissolved oxygen (DO); (d)-nitrate (N); (e)-phosphate (P); (f)-silicate (Si). Three statistics approaches used are the mean bias (MB), the root mean squared error (RMSE), and the coefficient of determination ($R^2$). N represents the number of data points.*

**6. In Figures 9 and 11, I am unable to understand what "ANN Model - RMSE" and "ANN Model + RMSE", and thus the related discussions.**

"ANN model + RMSE" and "ANN model - RMSE" represent upper and lower bounds of the ANN model accuracy. See Figure 10 (Line 533) and 12 (Line 541) of the revised manuscript.

**Referee comment #1 and #2**

**There are problems with low resolution for all of the figures. Figure 1 is not really even readable. Figures need to be re-generated with much higher resolution, or using vector, rather than raster, formats.                   From comment#1**

**Please improve the quality of figures 1 and 6, as they are difficult to read and follow.                   From comment#2**

We agree with the referees. Figures will be re-generated with higher resolution or using vector. As follows:

[revised manuscript text omitted]

**Figure S1 (revised): Comparison of three transfer functions. (a)-Log-sigmoid transfer function (logsig); (b) Hyperbolic tangent sigmoid transfer function (tansig); (c)-Positive linear transfer function (poslin).**

[Figure]

**Figure S2 (revised): Comparison of monthly-average environmental variables from the Changjiang Biology FVCOM with the corresponding observations at the surface and bottom on the East China Sea shelf. Blue and green solid lines represent surface and bottom simulated data from the Changjiang Biology FVCOM, respectively; red and black points show surface and bottom observation data from 2013 to 2016, respectively. (a)-temperature; (b)-salinity; (c)-dissolved oxygen; (d)-nitrate; (e)-phosphate; (f)-silicate.**

[Figure]

**Figure S3 (revised): Comparison of monthly average pH$_T$ on the East China Sea shelf. Blue solid line represents retrieved pH$_T$ by the ANN model using Changjiang Biology FVCOM output; green solid line represents simulated pH$_T$ by the Changjiang Biology FVCOM; red points show monthly average pH$_T$ observations from 2013-2016. (a)-surface; (b)-bottom.**

[Figure]

**Figure S4 (revised): Monthly average water discharge and its standard deviation (DaTong Station, data derived from the Hydrological Information Center of China, http://www.hydroinfo.gov.cn/).**

[Figure]

**Figure S5 (revised): Seasonal cycles of surface T (a), S (b), AOU (c), and P (d) from Changjiang Biology FVCOM output 
[revised manuscript text omitted]

**Supplementary material**

**Table S1: The performance of different number of neurons for two hidden layers in the training step. Three statistics are the coefficient of determination ($R^2$), the root mean squared error (RMSE), and the mean absolute error (MAE).**

| Model | Number of neurons | | Training data | | | Independent validation data | | |
|---|---|---|---|---|---|---|---|---|
| | first hidden | second hidden | $R^2$ | RMSE | MAE | $R^2$ | RMSE | MAE |
| 1 | 4 | 4 | 0.68 | 0.071 | 0.054 | 0.67 | 0.072 | 0.057 |
| 2 | 8 | 4 | 0.70 | 0.070 | 0.050 | 0.67 | 0.069 | 0.050 |
| 3 | 16 | 4 | 0.76 | 0.062 | 0.045 | 0.76 | 0.062 | 0.045 |
| 4 | 32 | 4 | 0.74 | 0.063 | 0.046 | 0.79 | 0.062 | 0.048 |
| 5 | 40 | 4 | 0.76 | 0.062 | 0.044 | 0.76 | 0.061 | 0.045 |
| 6 | 64 | 4 | 0.79 | 0.058 | 0.041 | 0.78 | 0.056 | 0.043 |
| 7 | 128 | 4 | 0.76 | 0.062 | 0.045 | 0.74 | 0.062 | 0.044 |
| 8 | 8 | 8 | 0.73 | 0.065 | 0.047 | 0.73 | 0.065 | 0.048 |
| 9 | 16 | 8 | 0.78 | 0.059 | 0.042 | 0.78 | 0.058 | 0.044 |
| 10 | 32 | 8 | 0.78 | 0.059 | 0.042 | 0.83 | 0.053 | 0.039 |
| 11 | 40 | 8 | 0.79 | 0.059 | 0.042 | 0.77 | 0.055 | 0.040 |
| 12 | 64 | 8 | 0.77 | 0.061 | 0.044 | 0.76 | 0.059 | 0.042 |
| 13 | 128 | 8 | 0.77 | 0.060 | 0.042 | 0.79 | 0.059 | 0.043 |
| 14 | 16 | 16 | 0.79 | 0.057 | 0.041 | 0.85 | 0.054 | 0.041 |
| 15 | 32 | 16 | 0.80 | 0.057 | 0.040 | 0.69 | 0.059 | 0.043 |
| 16 | 40 | 16 | 0.82 | 0.054 | 0.039 | 0.81 | 0.053 | 0.039 |
| 17 | 64 | 16 | 0.79 | 0.059 | 0.041 | 0.76 | 0.057 | 0.040 |
| 18 | 128 | 16 | 0.79 | 0.058 | 0.040 | 0.78 | 0.059 | 0.043 |
| 19 | 32 | 32 | 0.78 | 0.059 | 0.042 | 0.75 | 0.058 | 0.039 |
| 20 | 40 | 32 | 0.79 | 0.058 | 0.041 | 0.79 | 0.055 | 0.040 |
| 21 | 64 | 32 | 0.78 | 0.059 | 0.042 | 0.83 | 0.052 | 0.040 |
| 22 | 128 | 32 | 0.79 | 0.058 | 0.041 | 0.79 | 0.056 | 0.041 |
| 23 | 40 | 40 | 0.77 | 0.060 | 0.043 | 0.77 | 0.060 | 0.044 |
| 24 | 64 | 40 | 0.79 | 0.058 | 0.042 | 0.75 | 0.060 | 0.043 |
| 25 | 128 | 40 | 0.80 | 0.057 | 0.040 | 0.78 | 0.057 | 0.042 |
| 26 | 64 | 64 | 0.78 | 0.060 | 0.042 | 0.78 | 0.057 | 0.040 |
| 27 | 128 | 64 | 0.72 | 0.068 | 0.050 | 0.65 | 0.067 | 0.048 |
| 28 | 128 | 128 | 0.72 | 0.067 | 0.049 | 0.65 | 0.072 | 0.051 |

[Figure]

**Figure S1: Comparison of three transfer functions. (a)-Log-sigmoid transfer function (logsig); (b) Hyperbolic tangent sigmoid**
**transfer function (tansig); (c)-Positive linear transfer function (poslin).**

[Figure]

**Figure S2: Comparison of monthly-average environmental variables from the Changjiang Biology FVCOM with the corresponding**
**observations at the surface and bottom on the East China Sea shelf. Blue and green solid lines represent surface and bottom**
**simulated data from the Changjiang Biology FVCOM, respectively; red and black points show surface and bottom observation data**
**from 2013 to 2016, respectively. (a)-temperature; (b)-salinity; (c)-dissolved oxygen; (d)-nitrate; (e)-phosphate; (f)-silicate.**

[Figure]

**Figure S3: Comparison of monthly average pH$_T$ on the East China Sea shelf. Blue solid line represents retrieved pH$_T$ by the ANN**
**model using Changjiang Biology FVCOM output; green solid line represents simulated pH$_T$ from the Changjiang Biology FVCOM;**
**red points show monthly average pH$_T$ observations from 2013-2016. (a)-surface; (b)-bottom.**

[Figure]

**Figure S4: Monthly average water discharge and its standard deviation (DaTong Station, data derived from the Hydrological**
**Information Center of China, http://www.hydroinfo.gov.cn/).**

[Figure]

**Figure S5: Seasonal cycles of surface T (a), S (b), AOU (c), and P (d) from Changjiang Biology FVCOM output on the East China**
**Sea shelf from 2000-2016. The green circles represent monthly regional average, the blue dashed represents mean value of each**
**month.**